# Computer-Aided Diagnosis System of Alzheimer’s Disease Based on Multimodal Fusion: Tissue Quantification Based on the Hybrid Fuzzy-Genetic-Possibilistic Model and Discriminative Classification Based on the SVDD Model

**DOI:** 10.3390/brainsci9100289

**Published:** 2019-10-22

**Authors:** Lilia Lazli, Mounir Boukadoum, Otmane Ait Mohamed

**Affiliations:** 1Department of Electrical Engineering, École de Technologie Supérieure, ÉTS, University of Quebec, Montreal, QC H3C 1K3, Canada; 2CoFaMic Research Center, Computer Science Department, Université du Québec à Montréal, UQAM, University of Quebec, Montreal, QC H3C 3P8, Canada; mounirboukadoum@courrier.uqam.ca; 3Computer Science Department, Faculty of Engineering Sciences, University of Badji Mokhtar Annaba, UBMA, Annaba 23000, Algeria; 4Department of Electrical and Computer Engineering, Concordia University, Montreal, QC H3G 1M8, Canada; ait-mohamed@gmail.com

**Keywords:** Alzheimer’s disease, CAD system, multimodal fusion, tissue volume quantification, bias corrected FCM clustering, genetic optimization, possibilistic FCM clustering, SVDD classifier

## Abstract

An improved computer-aided diagnosis (CAD) system is proposed for the early diagnosis of Alzheimer’s disease (AD) based on the fusion of anatomical (magnetic resonance imaging (MRI)) and functional (^8^F-fluorodeoxyglucose positron emission tomography (FDG-PET)) multimodal images, and which helps to address the strong ambiguity or the uncertainty produced in brain images. The merit of this fusion is that it provides anatomical information for the accurate detection of pathological areas characterized in functional imaging by physiological abnormalities. First, quantification of brain tissue volumes is proposed based on a fusion scheme in three successive steps: modeling, fusion and decision. (1) Modeling which consists of three sub-steps: the initialization of the centroids of the tissue clusters by applying the Bias corrected Fuzzy C-Means (FCM) clustering algorithm. Then, the optimization of the initial partition is performed by running genetic algorithms. Finally, the creation of white matter (WM), gray matter (GM) and cerebrospinal fluid (CSF) tissue maps by applying the Possibilistic FCM clustering algorithm. (2) Fusion using a possibilistic operator to merge the maps of the MRI and PET images highlighting redundancies and managing ambiguities. (3) Decision offering more representative anatomo-functional fusion images. Second, a support vector data description (SVDD) classifier is used that must reliably distinguish AD from normal aging and automatically detects outliers. The “divide and conquer” strategy is then used, which speeds up the SVDD process and reduces the load and cost of the calculating. The robustness of the tissue quantification process is proven against noise (20% level), partial volume effects and when inhomogeneities of spatial intensity are high. Thus, the superiority of the SVDD classifier over competing conventional systems is also demonstrated with the adoption of the 10-fold cross-validation approach for synthetic datasets (Alzheimer disease neuroimaging (ADNI) and Open Access Series of Imaging Studies (OASIS)) and real images. The percentage of classification in terms of accuracy, sensitivity, specificity and area under ROC curve was 93.65%, 90.08%, 92.75% and 97.3%; 91.46%, 92%, 91.78% and 96.7%; 85.09%, 86.41%, 84.92% and 94.6% in the case of the ADNI, OASIS and real images respectively.

## 1. Introduction

According to “World Alzheimer Report 2018” [1] globally, there are 50 million patients suffering from dementia, plus a new patient is touched every three seconds with health care that costs 1 trillion US dollars. The “World Health Organisation” predicts that by 2050, 152 million people will be affected, and the costs will reach US $2 trillion by 2030. Alzheimer’s disease (AD) is the most dominant pathology present in 60% to 70% of cases of neurodegenerative dementia [1]. According to researchers from the “Canadian Outcomes Study in Dementia” [2], slowing the progression of this disease will contribute significantly to the reduction of its economic and psychosocial costs. This slowdown depends in particular on the implementation of early interventions, which are possible following an early diagnosis of the disease. 

Among the branches of neuroimaging, computer-aided diagnosis (CAD) is a powerful tool that provides early diagnosis of disease progression in a cost-effective and unbiased manner with respect to human inconsistencies. Although several works [3,4,5,6,7,8,9] have been done in the last decade by providing CAD systems for accurate diagnosis, however, few medical image processing tools have been developed to analyze the extensive amount of generated data and to assimilate the rather complex structures of the cerebral image. In this context, the collection of various data, resulting from various modalities: magnetic resonance imaging (MRI), single photon emission computed tomography (SPECT) and positron emission tomography (PET) as well as expert knowledge, becomes moreover more common in clinical departments for the study of Alzheimer’s pathology. The exploitation of all these data, performed by the clinician who analyzes and aggregates the data according to his knowledge, generally leads to a more precise, clearer and more reliable diagnosis. 

So, the main motivation for this work is to model the expert aggregation process using multimodal fusion techniques. This requires appropriate algorithms for getting a precise description of the regions of interest and the brain tissue affected by AD. Our aim is to use the high resolution anatomical information provided by MRI and the low resolution, highly functional information provided by PET to synthesize a high resolution functional image. The main interest of this type of fusion is to exploit the anatomical information for the precise detection of pathological zones characterized in functional imaging by physiological anomalies. Thus, to meet this multimodal fusion criterion, we propose a CAD system for AD early diagnosis based on MRI-PET multimodal fusion, powerful segmentation and classification approaches whose developed generic process is described hereinafter. 

The study of Xue Hua et al. [10] has shown that AD affects the brain by overwhelming tissue volumes of gray matter (GM), white matter (WM), and cerebrospinal fluid (CSF). It is therefore important to achieve a precise quantification of the volume of these tissues that would facilitate the correct diagnosis and understanding of this dementia. However, the task of precise segmentation and tissue volumes quantification is quite complex and one of the open problems in image processing. This is due to the fuzzy boundaries between the tissues particularly, GM and WM. Noting also the uncertainty that comes from noise, partial volume effect (PVE) and anatomical variations within “pure” tissue compartments and deformation in tissue volume resulting from AD and treatment. The purpose of our segmentation method is to assist the physician by first assigning the voxel fuzzy memberships, representing the possibility that they have to belong to brain tissue, as well as to calculate segmentation based on the fusion of relevant and redundant information of brain images. So, at first, we performed a segmentation based on fuzzy logic, especially on possibilistic theory, which has the potential to overcome the ambiguity and uncertainty of images [11,12]. The proposed fusion process is divided into three stages: (1) Modeling: Fuzzy tissue maps are first calculated for all anatomical and functional images. (2) The fusion is then performed for all the tissues with a context-based possibilistic fusion operator. (3) Decision: The final segmentation makes it possible to compute the volumetric measurement of brain tissue volumes and create a new synthetic anatomo-functional image.

In the modeling phase, a hybrid clustering algorithm is proposed; we adopt the possibilistic Fuzzy C-Means (PFCM) [13] which has a fuzzy and possibilistic membership at a time. Pal et al. [13] has shown that this algorithm can overcome the weakness of the FCM with respect to noise and PVE and also solve the problem of the possibilistic C-Means (PCM) coincident class. This makes it possible to interpret the complex nature of (GM/WM) or (GM/CSF) interfaces in brain images, where a large number of voxels contain a mixture of two or three tissues. 

For the initialization of the PFCM parameters, we used the bias corrected FCM algorithm (BCFCM) proposed in [14]. In the BCFCM algorithm, the objective function of the FCM is modified to solve the problem of inhomogeneous intensity of brain images. For this reason, we introduce labeling term for voxel in order to influence it by the intensity of the immediate voxel in the neighborhood. This district serves as a regularizer and directs the solution towards homogeneous piece-wise labeling. This is very useful in the case of segmentation of a cerebral image corrupted by noise [14]. We then propose a genetic optimization for the initialization of the centers of clusters. To this end, we use the genetic algorithms (GA) [15] to determine the appropriate cluster centers and the corresponding fuzzy partition matrix. This empirical initialization avoids local minima and allows the clustering process to converge quickly [16].

To achieve good performance, it is important to develop a robust CAD system that can model and correctly recognize a single class and/or solve multiclass problems in the presence of a non-exhaustive representation of the classes in the learning dataset. In the latter case, the system should be able to recognize the classes modeled in the learning set by correctly rejecting the samples belonging to other classes. We address this critical issue by considering classification system consisting of one-class classifiers, specialized in the identification of a given class and the rejection of others present in the scene. Then, in a second step of our CAD system, a classification is made on the basis of a compact discriminative model. We introduce a general framework for using support vector data description (SVDD) Models [17], which discriminates healthy subjects from subjects with AD.

The utility of this one-class that is based on the establishment of a kernel, proves in its effectiveness to detect outliers and its high accuracy. To add to this, the SVDD has proved its robustness in various applications related to different domains [18,19,20,21,22,23,24], in the case where no prior knowledge on the distribution of the data is available [25]. This is due to its criterion of taking into consideration only those samples that belong to the target class in order to train the underlying data distribution. Therefore, it is powerful in characterizing the class of interest by rejecting the rest of classes. Given that this methodology is based on the notion of kernel, it inherits the associated advantages [25]. To overcome the scale problem of the SVDD, we applied the “divide-and-conquer” strategy which allowed reducing the calculation cost. 

For the experiments, we used anatomical images (MRI) and functional images (^8^F-fluorodeoxyglucose (FDG)-PET), belonging to two synthetic databases: Alzheimer disease neuroimaging (ADNI) and open access Series of imaging studies (OASIS), as well the real images belonging to the Gabriel-Montpied hospital (GMPH) in Clermont ferrand, France. The efficiency of segmentation is evaluated visually by comparing the image of the final segmentation with that of the available ground truth image (images labeled by the expert). In addition, quantitative validity measures are used to evaluate the effectiveness of the proposed approaches for the CAD system: Tanimoto coefficients (TC) and Jaccard Similarity (JS) for segmentation approach and sensitivity (SE), specificity (SP), classification accuracy (CA) and area under ROC curve (AUC) for classification approach. In this context, the robustness of the tissue quantification process is proven against noise, PVE and when inhomogeneities of spatial intensity (ISI) are high. Thus, the superiority of the SVDD classifier over competing conventional systems is also demonstrated.

The rest of the paper is organized as follows: In Section 2, we present a review of some works in the field of neuroimaging related to automated segmentation of brain tissues and CAD systems, with a brief discussion. In Section 3, we summarize the principle of the proposed CAD system using possibilistic-fuzzy-genetic fusion scheme based segmentation and SVDD based classification. The experimental results on simulated and real data are presented in Section 4. Some concluding remarks are drawn in Section 5.

## 2. Related-Work

In recent decades, several studies have been proposed in the neuroimaging field, and relating to the AD diagnostic systems. In this context, various machine learning and intelligent technical tools have been successfully developed in the literature.

### 2.1. Related-Work to the MR Segmentation of Brain Regions

In the literature, a variety of works have preferred to use pattern recognition techniques (see [26] for more details), to segment brain images by magnetic resonance. Two types of approaches can be distinguished: supervised and unsupervised. The supervised approach, such as the statistical approach [27,28], the k- nearest neighbors [29,30,31] and artificial neural networks [32,33,34], needs a crisp label for some images. As a result, the images to be segmented are split into two sets, learning and testing. Therefore, the performance of this approach can be strongly related to the choice of learning set.

The unsupervised approach has a high repetition because its results are based in particular on the information of the image data itself. In addition, it almost dispenses with model assumption as well as the distribution of image data. Learning in this case is not necessary to segment the images. The principle consists in grouping the voxels in clusters with physical tissue labels, aiming to separate the characteristics in different clusters by minimizing an objective function [26]. There are two types of sub-approaches: hard and fuzzy clustering. The first aims to determine a crisp partition of a set of vectors in *C* clusters. Therefore, each voxel receives a unique tissue assignment.

On the other hand, a fuzzy clustering with the classic Bezdek FCM algorithm is widely used for MR image segmentation [35], because of the actual segmentation performance. It assigns a membership degree to each voxel, which indicates the voxel’s membership to a region (cluster). Indeed, voxels can be classified into several classes of tissues with a varying degree of belonging. This is one of the main motivations for using fuzzy clustering techniques in this study.

The scientific community is moving towards fuzzy logic approaches to model the complexity of medical data and to reinforce the most used algorithm, the FCM against noise and spatial inhomogeneities and PVE artifacts. Some related work is proposed in this section.

Wang et al. [36], propose the modified FCM algorithm that associates local neighborhood information with nonlocal in the clustering process to expand robustness against noise.

Shen et al. [37] proposed the improved FCM that modifies the voxel distance from class center, adding a neighborhood attraction, taking the intensity and distance criteria into consideration. 

Szilágyi et al. [38] proposed the enhanced FCM algorithm, which integrates a mean filter on the input image whose result is an average image. A linearly weighted sum is then calculated between the original image and the average image. 

Zhang and Chen [39] proposed the spatially constrained kernelized FCM algorithm that modifies the objective function of the FCM by integrating on the one hand, a distance based kernel-induced spatial constraint, and on the other hand, by incorporating a penalty term that contains spatial neighborhood information.

Chuang et al. [40] propose the FCMSI algorithm, which integrates into the membership value of each voxel, the sum of belongings to its neighborhoods. A control parameter that checks the trade-off between the original filtered images is usually estimated by trial and error experiments. 

Cai et al. [41] proposed the fast generalized FCM algorithm which exploits the local spatial and gray level relationships to specify a locality parameter that overrides the global parameter.

Karan Sikka et al. [42], propose the following process: a homomorphic filter based on entropy is used to correct the inhomogeneity. Then, a local peak merge method is proposed to initialize the centers of classes. The image voxels is finally grouped using a modified FCM which takes into account the neighborhood information. 

However, despite FCM and its variants being very useful classification methods, they cannot effectively compensate for intensity inhomogeneities and may be inaccurate in a noisy environment [43]. All these methods that modify the objective function lead to computational problems due to the modification of most equations as well as the objective function. In addition, degrees of membership do not always match those of the data. In FCM and its extensions, the generated membership values represent the degree of sharing, but not the degree of *typicality* or compatibility with elastic constraint. *Typicality* here means the degree to which a voxel belongs to a cluster rather than an arbitrary division of data. 

In the Krishnapuram and Keller [43] algorithm, the PCM is more robust to noise and PVE and approached this problem by associating absolute membership values *t_ij_* of the characteristics to clusters that represent the degree of *typicality* rather than the degree of sharing and hence the elimination of the FCM constraint ($\sum_{i=1}^{C}u_{ij}=1,j\in \left\{1,\ldots ,C\right\}$) is performed. Thus, each cluster is independent of other clusters in PCM. In this context, the PCM membership *t_ij_* of a voxel vector *x_j_* to tissue class *i* only depends on the distance *d*(*x_j_, b_i_*) between *x_j_* and class center *b_i_*, and not on the memberships of *x_j_* in all other classes, as is the case for FCM of which the analytic formulation of the membership degree *u_ij_* depends on the distances of *x_j_* to all class centers *b_k_*. This is very convenient in the case of strong ambiguity or uncertainty, which can occur in brain images. 

However, as pointed out by Barni et al. [44], PCM is highly dependent on the initialization and it sometimes generates coincident clusters. Furthermore, typicalities can be very sensitive to the choice of the additional parameters such as estimation of scale parameter $\eta_{i}$, which determines the distance at which the membership value of a voxel in a tissue cluster becomes 0.5. 

In addition, increasing the possible number of local minima can produce a number of bad minimizers likely to lock the PCM iterations into weak classification. The computational time required for PCM is *O*(*n*). It should be noted that PCM can generate trivial solutions since the solution spaces are not constant over all clusters; moreover, it only achieves a local minimum and so is unable to minimize the objective function in a global sense. 

The performance of PCM for noisy data can be improved using the possibilistic-fuzzy c-means (PFCM) clustering algorithm proposed by Pal and Bezdek, in [13]. It is a hybridization of PCM and FCM which solves the noise sensitivity defect of FCM and overcomes the coincident clusters problem of PCM. This is our motivation to choose, PFCM for the assessment of WM, GM and CSF volumes in the context of fusion scheme based tissue quantification. 

### 2.2. Related Work to Computer Aided-Diagnosis System of Alzheimer’s Disease 

Given the clinical accessibility of MRI clinically for neuroimaging, several studies have attempted to use images from synthetic bases such as ADNI and OASIS to exploit this anatomical modality or to associate it with various modalities using multimodal fusion techniques to improve the performance of CAD systems for AD.

In [45], Huang and Lee improved the performance of the maximization mutual information (MMI) approach by providing FCM/MMI fusion for MR and SPECT multimodal images to generate the fuzzy map of brain tissue. The error functions of brain slices were performed with three refined invariants: the area, the long axis and the short axis. The authors have published results for ADNI images that prove the speed and accuracy of the registration with the proposed fusion approach.

In [46], the authors propose an MR-PET multimodal CAD system based on the multiple-kernel learning (MKL) approach. The authors adopted the 10-fold cross-validation approach by publishing interesting results for ADNI database based on classification accuracy, sensitivity, and specificity.

In [47], the authors propose the hybrid Principal Component Analysis/Linear Discrimination Analysis (PCA/LDA) approach for extracting characteristics. The Fisher discriminant ratio (FDR) is then used to select the relevant characteristics. Two classifiers, the support vector machines (SVM) and the feed-forward neural network (FFNN) were used with PET images from the ADNI database. The results were based on classification accuracy, sensitivity, specificity and AUC (area under the ROC curve). The SVM outperformed the FFNN with better results.

In [48], the scale-invariant feature transforms (SIFT) approach is used for parameter extraction of MRI images from the OASIS database. A selection strategy was then used based on the application of Fisher’s discrimination report and GA. SVMs with different kernels were finally applied with "leave-one-out" cross-validation. 

In [49], Bhavana and Krishnappa improved the discrete wavelet transform (DWT) approach by proposing the intensity hue saturation (HIS) approach to merge PET and MRI ADNI images. The authors validated the results with four performance measures: mean squared error (MSE), peak signal-to-noise ratio (PSNR), average gradient (AG), and spectral discrepancy (SD) and with best visual observation. 

In [50], MKL method that combines the MRI, FDG-PET and CSF modalities are proposed to measure brain atrophy and to quantify hypo metabolism and proteins related to AD. The linear SVM classifier is then applied using 10-fold cross-validation. The published results for ADNI images showed good performance.

In [51], the sparse composite LDA (SCLDA) model is proposed to identify brain regions affected by early AD. Published results for ADNI multimodal images show good performance. 

In [52], a random forest-based fusion approach is proposed that combines regional MRI volumes, voxel-based FDG-PET signal intensities, CSF biomarker measurements, and categorical genetic information. Published results for ADNI images are vastly better than the exploitation of a single modality.

In [53], a fusion method that combines the MRI and PET ADNI images is proposed. A multi-kernel SVM is then applied for classification. The published results using classification accuracy, sensitivity, specificity and AUC demonstrate good results. 

In [54], multilevel convolutional neural networks (CNN) is proposed to train and combine the parameters of MRI and PET multimodal images from the ADNI database. Good performance is achieved in term of classification accuracy, sensitivity, specificity and AUC. 

In [6], a very deep convolution network is proposed, applying it on MRI images from the OASIS database. The performance of the approach is demonstrated in terms of classification accuracy with five-fold cross validation.

In a previous work [5], a hybrid FCM/ PCM segmentation process is proposed to evaluate the tissue volume of MR and PET images with 20% noise level from ADNI database. SVMs (with RBF kernel) were then used for classification. The performance with “leave-one-out” cross-validation strategy, and using classification accuracy, sensitivity and specificity measures was good.

### 2.3. Discussion Related to CAD Systems of Alzheimer’s Disease

The real challenge of CAD systems that are widely used in the clinical medical environment is to exploit new technologies capable of analyzing a large amount of data. Multimodal imaging is gaining more and more importance among these recent technologies. In this context, recent studies have attempted to simultaneously combine at least two or more biomarkers among MRI, PET, and CSF for the early diagnosis of AD. In contrast, research is still centered on a clinical core of early and important episodic memory disorders. In addition, the majority of systems apply for image registration purposes with the objective of helping physicians to locate the same region of interest on multiple images of the same patient. However, beyond the registration stage, many things remain to be done! The multimodal fusion idea that allows combining several images from different types of modalities, for the purpose of constructing a fusion image, is an interesting concept that opens several perspectives. Such a process could ideally suspend any unnecessary information found in the images taken separately. It thus allows combining the relevant data so that the new merged image is legible and easier for the physician to appreciate. In this fusion image, the physician could even visualize information that was not clear in both images. It is the goal of our CAD system to guide the physician with a first assignment by an automated system for AD diagnosis based on the SVDD classifier and a fuzzy-possibilistic-genetic fusion based segmentation, which provides relevant information from MRI/FDG-PET multimodal images.

## 3. Study on Patients with Alzheimer’s Disease: Method and Experience

The following is a description of an improved CAD system (Figure 1), allowing diagnosis of AD. 

### 3.1. Preprocessing and Registration

To delete non-brain tissue from an image, we have used the FSL-BET (*Brain Extraction* Tool) [55]. To partially correct the ISI, we used the neuroimaging Informatics *Tools*, especially the non-parametric model in the SPM8 (statistical parametric mapping) tool [56] with a minimizing function based on the entropy of the image intensity histogram. To address the non-uniform intensity problem, we adapted the N3 (Non-parametric Non-uniform intensity Normalization) method [57], and normalization of intensity values to ensure the same dynamic range values 0 to 1 for all images. 

Then we added white Gaussian noise with different signal-to-noise ratio (SNR) to quadrature components of the received signal to mimic the effect of many random processes. The hybrid median filter was applied after on the images. For each slice of MRI and PET images, we created several volumes (vol_1_, …, vol_60_), by varying the SNR additive noise levels from 1% to 20%, the slice thickness (ST) of 1, 3 and 5 mm without considering the space between the slices. The RF (inhomogeneity of the radio frequency) was chosen at 20% intensity non-uniformity. In addition, various regions have been formed for each slice whose gray level in each varies according to certain constraints. Each voxel *v* consists of a vector of parameters corresponding to the values of intensity (gray level) in these different regions.

We used the “imregister” function available in image processing toolbox of MATLAB R2018b for automatically aligning MRI and PET images to a common coordinate system using intensity-based image registration. Each voxel *v* is presented with Talairach x-y-z coordinates. The Volume Occupancy Talairach Labels (VOTL) database available through Freesurfer processing uses a volume-filling hierarchical naming scheme to organize labels for brain structures ranging from hemispheres to cytoarchitectural regions. The Talairach label data are content in an NIfTI image (Talairach.nii). From these files, the voxel coordinates are recorded in a 4 × 4 *sform* matrix, (1,1,1) are the coordinates of the first voxel at the bottom-left-posterior. Furthermore, the scaling is done by centering the brain on the AC (anterior commissure) point in the inter-hemispheric plane which will have for coordinates (0,0,0), then by cutting the brain in 12 rectangular boxes located on each side of the sagittal plane (x,z) and the axial plane (x,y) as well as between the two coronal planes (y,z) passing through the superior edge of AC and the inferior edge of PC ( posterior commissure). 

Each box is then deformed so that: PC has coordinates (0, −23, 0); the most anterior point of the brain has for coordinate y = 70; the most posterior point has for coordinate y = −102; the point of the most lateral x-axis to the right has at coordinate x = 68 and to the left x = −68; the highest point of the brain is at the z = 74 coordinate; the lowest point of the brain is at the z = −42 coordinate.

### 3.2. Segmentation

We have naturally built the segmentation module (Figure 2) based on fusion scheme in three successive steps: modeling, fusion and decision. 

#### 3.2.1. Modeling

It consists of three sub-steps: first, the initialization of the centroids of the tissue clusters by applying the BCFCM algorithm. Then, optimization of the initial partition is performed by running the GA. Finally, the creation of WM, GM and CSF fuzzy tissue maps by applying the PFCM algorithm.

##### Initialization by the Bias Corrected Fuzzy C-Means Algorithm

Due to the ISI and noise introduced in imaging process, different tissues at different locations may have a similar intensity appearance, whereas the same tissue at different locations may have a different intensity. As a result, the segmentation result will be improved after incorporating the spatial information. We propose to apply the BCFCM [14] algorithm to correct ISI.

A. Principle of the Bias Corrected Fuzzy C-Means algorithm: The modified objective function is given as:(1)$$J_{BCFCM}\left(B,U,X,\beta \right)=\overset{C}{\underset{i=1}{\sum }}\overset{N}{\underset{j=1}{\sum }}u_{ij}^{m}\|x_{j}-\beta_{j}-b_{i}{\|}^{2}+\frac{\alpha }{N_{i}}\overset{C}{\underset{i=1}{\sum }}\overset{N}{\underset{j=1}{\sum }}u_{ij}^{m}\left(\underset{x_{k}\in N\left(x_{j}\right)}{\sum }\|x_{k}-\beta_{k}-b_{i}{\|}^{2}\right)$$
where:B, the matrice of centroids and $b_{i},$ the center of cluster *i* (1 ≤ *i* ≤ *C*) with *C,* the number of cluster.*X,* the matrice of voxels vectors and *x_j_* (1 ≤ *j* ≤ *N*), the observed log-transformed intensities at the *j^th^* voxel. U, the matrice of degrees of membership [$\mu_{ij}^{m}$] with *m,* a parameter controlling the degree of fuzzification. *β_j,_* the bias field value at the *j*^th^ voxel, that helps in removing the inhomogeneity effect. *N_i_* represents size of neighborhood to be considered. The neighborhood effect is controlled by the parameter *α* whose selection strongly affects the precision of the results.$N\left(x_{j}\right)$ stands for set of neighbours that exists in a window around *x_j_* and is the cardinality of *N_i_*.

The effect of the neighbors’ term is controlled by parameter α. The relative importance of the regularizing term is inversely proportional to SNR of the brain signal. Lower SNR would require a higher value of the parameter α. The objective function *J_BCFCM_* (Equation (1)) is minimized under the following constraints:(2)$$U\left\{u_{ij}\in \left[0,1\right]|\sum_{i=1}^{C}u_{ij}=1\forall j\;and\;0<\sum_{j=1}^{N}u_{ij}<N\forall i\right\}$$

The membership partition matrix, cluster centroid and bias field estimators are updated respectively as follows: (3)$$u_{ij}^{*}=\frac{1}{\sum_{k=1}^{C}{\left(\left(w_{ij}+\left(\alpha /N_{i}\right)\gamma_{i}\right)/\left(w_{kj}+\left(\alpha /N_{i}\right)\gamma_{k}\right)\right)}^{1/\left(m-1\right)}}$$
where:$w_{ij}=\|x_{j}-\beta_{j}-b_{i}{\|}^{2}$$\gamma_{i}=\underset{x_{k}\in N\left(x_{j}\right)}{\sum }\|x_{k}-\beta_{j}-b_{i}{\|}^{2}$(4)$$b_{i}^{*}=\frac{\sum_{j=1}^{N}u_{ij}^{m}\left(\left(x_{j}-\beta_{j}\right)+\left(\alpha /N_{i}\right)\sum_{x_{k}\in N\left(x_{j}\right)}\left(x_{k}-\beta_{j}\right)\right)}{\left(1+\alpha \right)\sum_{j=1}^{N}u_{ij}^{m}}$$(5)$$\beta_{j}^{*}=x_{j}-\frac{\sum_{i=1}^{C}u_{ij}^{m}b_{i}}{\sum_{i=1}^{C}u_{ij}^{m}}$$

B. Application of BFCM algorithm: The process is applied to slices to initialize the centroids *b_i_* of tissue clusters and the corresponding fuzzy partition matrix which contains the initial degree of membership of each voxel to tissue. The BCFCM follows the steps of Algorithm 1. Applying the BCFCM, the anatomical and functional models consisted of a set of fuzzy tissue membership volumes, one for each brain tissue class. In these volumes, the voxel values reflected the proportion of tissue present in that voxel. 


**Algorithm 1: BCFCM Pseudo-Code**
Let $X=\left\{x_{j}\right\}$ the voxels set, $U=\left\{\mu_{ij}\right\}$ the matrix of membership degrees and $B=\left\{b_{ij}\right\}$ the matrix of cluster centers with 1≤i≤C,1≤j≤N.
*m* the degree of fuzzy and ε the threshold representing convergence error.Initialize the centers vectors *B*^(0)^ = [*b*_j_] and the degrees of belonging matrix *U*^(0)^ by random values in the interval [0, 1] satisfying the Equation (2).At k-step:-Compute the belonging degrees matrix *U*^(k)^ using Equation (3).-Compute the centers vectors *B*^(*k*)^ = [*b*_j_] using Equation (4).-Estimate the bias term $\beta_{j}^{\left(k\right)}$ using Equation (5).-Compute the objective function $J_{BCFCM}^{\left(k\right)}$ using Equation (1).Update: *B*^(*k+1*)^, *U*^(*k+1*)^, $\beta_{j}^{\left(k+1\right)}$ and $J_{BCFCM}^{\left(k+1\right)}$If $\|J_{BCFCM}^{\left(k+1\right)}-J_{BCFCM}^{\left(k\right)}\|<\varepsilon$ then STOP otherwise return to step 2.

##### Optimization by the Genetic Algorithms

The result of BCFCM clustering was used as initial population for GA which allows training the GA with a population of empirically generated chromosomes and not randomly initialized, avoiding the problem of local optima. 

The GA principle starts by initializing a population consisting of a set of chromosomes. For this purpose, the BFCM algorithm is applied in several attempts to generate these chromosomes in order to obtain an initial population. By executing GA operators (crossover and mutation), this population is used to create new populations in order to achieve a better one (offspring). The chromosomes are chosen according to their fitness, the more appropriate they are, the more chances they are to be exploited for reproduction. The following merit function was used to evaluate the fitness of a chromosome:(6)$$w=\sum_{l=1}^{M}\sum_{x_{i}\in C_{l}}p_{i}d^{2}(x_{i},g_{l})$$
where p_i_ is the weight of the i^th^ voxel and $g_{l}$ the center of gravity of cluster *C_l_*, *l* referring to one of the *M* clusters. We have:(7)$$g_{l}^{}=\frac{1}{\left|C_{l}\right|}\sum_{x_{i}\in C_{l}}x_{i}^{}$$

We varied the number of chromosomes from two to 20, the good result of the segmentation increased progressively before stabilizing for 10 chromosomes obtained by applying BCFCM in several experiments. We followed the suggestion proposed in [15] to adjust the GA parameters, then a value ≥ 0.5 is chosen for the probability of crossover *P_c_* and a value inversely proportional to the size of the population is adapted for the probability of mutation *P_m_*. Since the fitness function reached a minimum between 15 and 20 iterations, we used the latter value as a stop criterion and the chromosome with the lowest fitness value as an input for the modelization step. The GA parameters were set as follows: population size Ω = 10, stopping criterion = 20 iterations, probability of mutation *P_m_* = 0.01 and probability of crossover *P_c_* = 0.8.

##### Modelization by Possibilistic Fuzzy C-Means Algorithm 

We clustered the brain images with PFCM algorithm for which BCFCM/AG process has been used to choose the initial partition (fuzzy memberships degrees to tissues and tissue centers). 

A. Principle of Possibilistic fuzzy c-means algorithm: The advantage of the PFCM clustering [13] is that it generates the two membership functions simultaneously of the PCM and FCM algorithms. The first function represents the possibilistic membership (*t*_*ij*_) of the absolute degrees of typicality, and the second characterizes the fuzzy membership (*u*_*ij*_) of the relative degrees. This mixed algorithm makes it possible to associate with each feature vector of the voxel *x_i_*, *t*_*ij*_ and *u*_*ij*_ in each of the *C* clusters, reflecting the degree of belonging of the voxel *x_i_*, to cluster *j*. Iterative optimization is applied to approach the minima of a PFCM constrained objective function:(8)$$J_{PFCM}\left(U,B,X,m,\eta \right)=\overset{N}{\underset{i=1}{\sum }}\overset{C}{\underset{j=1}{\sum }}\left(au_{ij}^{m}+bt_{ij}^{\eta }\right)\|x_{i}-b_{j}{\|}^{2}+\overset{C}{\underset{j=1}{\sum }}\gamma_{j}\overset{N}{\underset{i=1}{\sum }}{\left(1-t_{ij}\right)}^{\eta }$$
where *u*_*ij*_ are constrained by the probabilistic conditions, while $t_{ij}$∈ [0,1] are subject to:$$0<\overset{C}{\underset{i=1}{\sum }}t_{ij}<C,\forall j$$

The variables $\left(\gamma_{i}>0,i=1,\ldots ,C\right)$ are user defined constants, which represent the possibilistic penalty terms and control the variance of the clusters. The fuzzy exponent *m* and possibilistic exponent *η* must be greater than 1, while *a* (*a* > 0) and *b* (*b* > 0) are tradeoff parameters to set the balance between the probabilistic and possibilistic term. The parameters *a* and *b* define the relative importance of membership and *typicality* in the computation of centroids. 

Unlike most fuzzy and possibilistic clustering algorithms, PFCM produces three outputs: fuzzy partition or membership matrix *U*, possibility matrix *T* of typicalities and a set *B* of *C* prototypes *b_i_* that compactly represents the clusters. So, *J*_PFCM_ can be minimized only if:(9)$$u_{ij}={\left(\overset{C}{\underset{k=1}{\sum }}{\left(\frac{\|x_{j}-b_{i}{\|}^{2}}{\|x_{j}-b_{k}{\|}^{2}}\right)}^{1/m-1}\right)}^{-1}1\leq j\leq N,1\leq i\leq C$$
(10)$$t_{ij}=\frac{1}{\left(1+{\left(\frac{b\|x_{j}-b_{i}{\|}^{2}}{\gamma_{i}}\right)}^{1/\eta -1}\right)}\forall i=1\ldots C,\forall j=1\ldots N$$
(11)$$b_{i}=\overset{N}{\underset{j=1}{\sum }}\left(\left(au_{ij}^{m}+bt_{ij}^{\eta }\right)x_{j}\right)/\left(au_{ij}^{m}+bt_{ij}^{\eta }\right)\forall i=1\ldots C$$

B. Application of PFCM algorithm: The pseudo-code of the PFCM clustering is described in Algorithm 2. The PFCM parameters were set as follows: *ε* = 0.005, *m* ∈ [1.5, 3], and η ∈ [3,5]. The Euclidean distance was applied, because it is the most classical distance and the least restrictive.


**Algorithm 2: PFCM Pseudo-Code**
Let $X=\left\{x_{j}\right\}$ the vectors of voxels, $U=\left\{\mu_{ij}\right\}$ the matrix of membership degrees, $T=\left\{t_{ij}\right\}$ the matrix of typicality degrees, $B=\left\{b_{ij}\right\}$ the matrix of cluster centers with 1≤i≤C,1≤j≤N.
*m* the degree of fuzzy and η the degree of weight possibilistic. Initialize the centers vectors *B*^(0)^ = [*b*_j_] and the degrees of belonging matrix *U*^(0)^ using hybrid BCFCM-GA method.At k-step:-Compute the matrix of membership degrees *U*^(k)^ using Equation (9).-Compute the matrix of typicality degrees *T*^(k)^ using Equation (10).-Compute the prototype matrix *B*^(*k*)^ using Equation (11).-Compute the objective function $J_{\mathrm{PFCM}}^{\left(k\right)}$ using Equation (8).Update: *U*^(*k+1*)^, *T*^(*k+1*)^, *B*^(*k+1*)^ and $J_{\mathrm{PFCM}}^{\left(k+1\right)}$Repeat steps (2) and (3) until the stop criterion is met: $\|J_{\mathrm{PFCM}}^{\left(k+1\right)}-J_{\mathrm{PFCM}}^{\left(k\right)}\|<\varepsilon$

There is no theoretical process to define the optimal value of *m*. If the latter is close to 1, the partition produced by PFCM is almost crisp, so voxel fuzzy memberships become more fuzzy as the value of *m* increases. When *m* tends to infinity, all memberships are equal to 1 / number of clusters *C* [58]. To guarantee the convergence, we followed the suggestions proposed in [58] by forming the PFCM algorithm with values of *m* belonging to the interval [1,5], which gave us a "better" partition of brain images. After some experiments, the value of *m* = 2 was maintained which leads also to the best SVDD classification accuracy (see Section 4.3 for the results).

The result obtained from the execution of the modeling process to the image *k* ∈ {1..*P*} is a series of three fuzzy maps corresponding to the tissue T∈ {CSF, GM, WM} estimated from the image *k*. Figure 3 illustrates the results of some slices which are chosen because of their tissue particularities. In this figure and in each map, the WM, GM and CSF tissues are expressed by the zones with white color. These fuzzy maps are related to three distributions of possibility $\pi_{i}^{T}$ ($i=\bar{1..N}\mathrm{voxels}$), where the value $\pi_{i}^{T}\left(v\right)$ is the membership of voxel *v* to tissue *T* computed from image *k*. Two sets of fuzzy maps are obtained for MRI and PET images: {WM_MRI_, GM_MRI_, CSF_MRI_}, {WM_PETI_, GM_PET_, CSF_PET_}. These maps are used in the next step to obtain fusion maps.

#### 3.2.2. Fusion 

We chose for the anatomical/functional fusion a possibilistic operator based on the conjunctive context which was introduced in [11], and which made it possible to take into account the ambiguous but relevant areas for the diagnosis. 

If ($\pi_{t}^{1},\pi_{t}^{2}$) are the gray-levels possibility distributions of a voxel *v* to tissue *T* derived from T_MRI_ and T_PET_ fuzzy maps, resulting from the modeling step, we applied a fusion operator *FOP* which computes a new membership value $\pi_{t}^{fus}$ according to the existing ambiguity and the redundancy between the two fuzzy maps. 

In another sense, for a pair of images (MRI, PET) and a given tissue *T* (GM, WM, CSF), the chosen fusion operator should be efficient to combine the possibilistic distributions for all the voxels of the two types of images highlighting redundancies and managing ambiguities and complementarities. The following few operators have been chosen for this operation [11]:(12)$$FOP_{1}:\pi_{t}^{fus}\left(v\right)=max\left(\frac{min\left(\pi_{t}^{1}\left(v\right),\pi_{t}^{2}\left(v\right)\right)}{h},1-h\right)$$
(13)$$FOP_{2}:\pi_{t}^{fus}\left(v\right)=min\left(1,\frac{min\left(\pi_{t}^{1}\left(v\right),\pi_{t}^{2}\left(v\right)\right)}{h}+1-h\right)$$
(14)$$FOP_{3}:\pi_{t}^{fus}\left(v\right)=min\left(\pi_{t}^{1}\left(v\right),\pi_{t}^{2}\left(v\right)\right)+1-h$$
(15)$$FOP_{4}:\pi_{t}^{fus}\left(v\right)=max\left(\frac{min\left(\pi_{t}^{1}\left(v\right),\pi_{t}^{2}\left(v\right)\right)}{h},min\left(max\left(\pi_{t}^{1}\left(v\right),\pi_{t}^{2}\left(v\right)\right),1-h\right)\right)$$
where:(16)$$h=1-\underset{v\epsilon Image}{\sum }\left|\pi_{t}^{1}\left(v\right)-\pi_{t}^{2}\left(v\right)\right|/\left|Image\right|$$

The quantity *h* is a measure of agreement between the two distributions of possibility (average distance between the two fuzzy maps of the same tissue *T*). The behavior of the combination operators *FOP*_i_ (*i* = 1…4) is quite similar and obeys a certain construction logic, illustrated here on the operator *FOP*_4_. If the two images are reliable, then a renormalized T-norm $(min\left(\pi_{T}^{1}\left(v\right),\pi_{T}^{2}\left(v\right)\right)/h$) is used for a conjunctive combination. In the opposite case, it is assumed that at least one of the two images is reliable and the operator acts cautiously.

With respect to CSF tissue, we have privileged the anatomical information for this tissue, and assigned to the corresponding fused map the fuzzy anatomical segmentation resulting from the modeling step, due to the impertinence of the information resulting from the functional image.

#### 3.2.3. Decision

In this step, we pass from information provided by the sources to the choice of a decision which synthesizes the available information, by creating both a labeled and a synthetic image. They represent the final result of the automatic segmentation process.

##### Image Labeling

A segmented image was finally computed using the fuzzy maps obtained in the modeling step. Any voxel *v* in the segmented image was assigned to the label (GM, WM, CSF tissue) for which it had the highest degree of membership applying the maximum possibility rule [12]:(17)$$if\;T,T^{\prime }=\left\{GM,WM,CSF\right\}\;and\;T^{\prime }\neq T;v\in T\;if\;\pi^{T}\left(v\right)>\pi^{T^{\prime }}\left(v\right)$$

An example presented in Figure 4, illustrating an anatomical image labeled WM, GM and CSF, resulting from automatic segmentation based on the proposed hybrid approach. The brain tissues in the calculated image are labeled WM (green color), GM (purple color) and CSF (blue color), where the tissue volumes are well defined and classes are clearly legible, with a clearer anatomical localization that allows localization of basal ganglia and cortex (GM), fissures and ventricles (CSF) and WM tissue.

##### Synthetic Image

The degrees of merged possibilities $\pi_{T}$ are considered as percentages of partial volume $p_{T}$. It seems intuitively satisfying to consider that the more a voxel *v* has the possibility of belonging to a class of tissue *T* (i.e., more $\pi_{T}$(*v*) is high), the more the tissue *T* must be present in this voxel. The percentages of partial volume $p_{T}$ of the tissue *T* are then calculated after normalization of the degrees of possibility [11] as follows:(18)$$\forall v\in Image,p_{T}\left(v\right)=\frac{\pi_{T}\left(v\right)}{\sum_{i=1}^{T}\pi_{i}\left(v\right)}$$

From the values of mean functional activity *b_i_* (centroids of the tissue classes) resulting from the classification by the PFCM algorithm, the intensity of each voxel *v* of the synthesis image is then calculated using the volume percentages, calculated by Equation (18), as follows:(19)$$v_{Synthesis\;image}=\overset{T}{\underset{i=1}{\sum }}b_{i}p_{i}\left(v\right)$$

For multimodal images visualization, we used Color Overlay fusion technique, available through the “Fusion Options” drop down menu of Fusion Viewer software tool [59]. 

An example is shown in Figure 5 of anatomical and functional images of the same brain as well as the merged image resulting from the two images. The synthesis image makes it possible to locate hypoperfused zones with the anatomical precision of the MR image. A great interest of this type of image is that it immediately makes it possible to distinguish between a really hyper-fused zone (low membership to GM in PET and strong membership to GM in MRI) and an expanded sulci (low adhesion to GM in PET and very strong membership to CSF in MRI). One can appreciate a high anatomical definition, visible at the level of the cortical sulci and the central nucleus, as well as a perfusion pattern similar to that of the PET images.

### 3.3. Classification

The synthetic images from the segmentation step are used in the classification step based on the application of the discriminant one-class SVDD and the “divide-and-conquer” strategy. 

#### 3.3.1. Principle of Operation of the SVDD

SVDD [17] is a machine learning approach that draws on the concept of support vector machines (SVM) for one-class classification. It is robust in locating outliers thanks to its principle based on the idea of constructing a hypersphere (*R*, *a*) with a minimum volume containing most of the target data, where *a* is the center and *R* is the radius of the minimal hypersphere. The points in the sphere are the support vectors. The objective or error function to be minimized is as follows:(20)$$\underset{R,\;a,\;\xi \;}{\min}F\left(R,a\right)=\underset{R,\;a,\xi \;}{\min}R^{2}+C\overset{N}{\underset{i=1}{\sum }}\xi_{i}$$

The objective function (Equation (20)) is reduced using the following constraint: (21)‖xi−a‖2≤R2+σ+ξi with ξi≥0, ∀i=1,…,N
where xi∈ℝm,i=1,…,N represents the training data with *N* is the total number of target data. σ is a width parameter of Gaussian RBF kernel function which is the only one to adjust to avoid numerical complexity [5,17]. *C* is a parameter that controls the tradeoff between volume of a hypersphere and the number of errors. ξi are the "slack" variables, which are introduced to allow the target points to get out of the sphere, and reduce the effect of outliers. 

We use the Lagrange multipliers below to integrate the previous constraint into the cost function:(22)$$L\left(R,a,\alpha_{i},\gamma_{i},\xi_{i}\right)=R^{2}+C\underset{i}{\sum }\xi_{i}-\underset{i}{\sum }\alpha_{i}\left(R^{2}+\xi_{i}-\left(\|x_{i}{\|}^{2}-2ax_{i}+\|a{\|}^{2}\right)\right)-\underset{i}{\sum }\gamma_{i}\xi_{i},\;\alpha_{i}\geq 0,\gamma_{i}\geq 0$$
where, L must be minimized with respect to *R*, *a*, $\xi_{i}$ and maximized with respect to $\alpha_{i}$ and $\gamma_{i}$. 

The optimal solution obtained should guarantee the conditions of KKT (Karush-Kuhn-Tucker), respecting the inequalities: $\alpha_{i}\geq 0$, $\gamma_{i}\geq 0$. 

To test if an image z is in the hyper-sphere, the distance to the center of the hyper-sphere is applied as follows:(23)$$\|z-a{\|}^{2}=K\left(z,z\right)-2\underset{i}{\sum }\alpha_{i}K\left(z,x_{i}\right)+\underset{i,j}{\sum }\alpha_{i}\alpha_{j}K\left(x_{i},x_{j}\right)\leq R^{2}$$

When the distance is equal to or less than the radius *R*, the test image *z* is accepted or inserted into the target class.

One of the fundamental problems when training the SVDD model is to find the optimal value of the Gaussian kernel parameter σ and the penalty factor *C*, in order to guarantee the best performance of the classification system. To perform hyper-parameter optimization, the grid search method is used to find the best combination of (σ, *C*). We have adjusted these two parameters in many experiments, finally choosing the best 18 combinations, we select then (0.5, 0.0040), (0.5, 0.0625) and (0.5, 0.0500) as optimal values for ADNI, OASIS and GMPH respectively (see Section 4.3 for SVDD results). Figure 6 presents some solutions for different values of *C* and σ.

#### 3.3.2. Adoption of the “Divide-and-Conquer” Strategy 

The widely used quadratic programming (QP) solver is replaced by the “divide and conquer” strategy to overcome the SVDD scale problem. So, we divide the dataset into a series of small subgroups or data classes by applying the k-means clustering algorithm. Each sub-problem is solved by its local expert with SVDD, and during final learning, we only use the support vectors (SV) that are found by each local expert, which allows the process to reduce the load and cost of the calculating and converging quickly. The process of division and training of this process referred to k-Means/SVDD is summarized in Figure 7.

#### 3.3.3. Computational Complexity

A QP solver runs in *O (N^3^)*, where *N* is the size of the training data. By reducing the size of the training data for the QP solver, the complexity of the k-means/SVDD model with *k* clusters is then: (24)$$O\left(kN\right)+kO\left({\left(N/k\right)}^{3}\right)+O\left({\left(\alpha k\right)}^{3}\right)\approx kO\left({\left(N/k\right)}^{3}\right)$$
where: *α* is the average number of SV for each sub-problem description.

Thus, the size of the work set is *αk*. The first term *O (kN)* represents the complexity of the k-means grouping algorithm. The second characterizes the complexity of the QP solver of the local SVDD, provided that the training data *N* is also distributed in *k* sub-problems. The third term *O((αk)^3^)* is for recycling. The size of the sets *αk* is smaller than the initial size of the training data *N*. Thus, because of the grouping and recycling steps, the complexity is rather low due to the decomposition effect.

## 4. Material, Quantitative Validation & Discussion

### 4.1. Information on Patients, Imaging Parameters and Acquisition 

The experiments are performed on the datasets described in Table 1.

ADNI [60]—1.5 Tesla scanner was used to acquire the MR images. Each slice of MR volume contains 256 × 256 × 176 voxels spanning the entire brain region, with the following parameters: the voxel size is 2 × 2 × 2 mm^3^. The isotropic resolution is 1.0 mm. The time of repetition (TR) is 5050 ms, and the time of echo (TE) is 10 ms. The acquisition of FDG-PET images was started after injection of the patient by tracing FDG for 30 to 60 min. The images were averaged, spatially aligned, interpolated to a standard voxel size, normalized in intensity, and smoothed to a common resolution of 8 mm wide at mid-height. Each slice of the reconstructed PET images has a size of 256 × 256 × 207 voxels and a voxel size of 1.2 × 1.2 × 1.2 mm^3^.

OASIS [61,62]—The MRI data was collected on the Siemens Vision 1.5T scanner (Siemens, Erlangen Germany). Participants receiving a simultaneous PET acquisition, on the BioGraph mMR, received a tracer injection prior to initiation of the MRI scan. The PET data were collected on the Siemens ECAT HRplus 962 PET scanner. Metabolic imaging with [18F] FDG-PET was performed with 3D dynamic acquisition started 40 min after a bolus injection of approximately 5 mCi FDG and lasted 20 min.

GMPH—A Magnetom 1-T imager (Siemens, Erlangen, Germany) was used to acquire the MR images, with a 1T superconducting magnet and a lead coil. The parameters TR and TE are respectively 2600 ms and 3.0 ms with an angle of turn of 35°, providing T1-weighted images with a Pelc angle (24). Each slice includes 256 × 256 × 176 voxels with a voxel size of 1 × 3.1 × 1 mm^3^ and providing an isotropic resolution of 1.0 mm. The PET images underwent a transmission scan of 6 minutes before the emission scan. Then, a 20 min emission scan was started immediately after injection of the 555 Mq [11C]-PIB tracer. Each slice of the reconstructed images has a size of 256 × 256 × 207 voxels with isotropic voxels of size 2 × 2 × 2 mm^3^ each. 

### 4.2. Performance Evaluation of the Segmentation

To assess the segmentation method, the segmented image obtained with the modeling approach is compared with the reference labeled image. The ground truth was available and the expert labeling of the tissue regions was known. The reference images of the three datasets are accompanied by volumetric segmentation files produced through Freesurfer processing. For GMPH images the manual segmentation was performed by one qualified expert from the hospital of Gabriel-Montpied, whereas one could refer to the following references [61,62,63] for more details on the complete list of researchers and experts for ADNI and OASIS. 

The quantitative evaluation is desirable for objective comparison because it is difficult to determine visually the difference between the results of the proposed approaches. The comparison with the result of atlas-based segmentation is done based on two segmentation metrics, namely TC, and JS which allow validating the performance of the segmentation approach.

*TC* was defined for a given tissue as the number of voxels affected simultaneously to the same tissue in both computed and ground truth images divided by the number of voxels assigned to the tissue in the two labeled images.

The *TC* index is close to 1 for very similar results and is near 0 when the labeled images share no similarly classified voxels. It is described as follows [58]:(25)$$TC\left(T\right)=\frac{v_{AR}^{T}}{v_{A}^{T}+v_{R}^{T}-v_{AR}^{T}}$$
where $v_{AR}^{T}$ is the number of voxels that are assigned to tissue *T* in automated labeled image (calculated by PFCM segmentation algorithm) and the reference labeled image (ground truth), $v_{A}^{T}$ and $v_{R}^{T}$ denote the number of voxels assigned to tissue *T* by the automated algorithm and ground truth respectively.

JS metric is used to evaluate the spatial overlap between the ground truth (*G*) and segmented (*S*) tissues. It is computed as the ratio between the intersection and union of the *S* and *G* tissues as follows:(26)$$JS=\frac{\left|S\cap G\right|}{\left|S\cup G\right|}$$

The JS values are between 0 and 1, greater values indicate that more voxels of the segmented tissue correspond to that of the ground truth image.

The performance of the proposed PFCM modeling with hybrid BCFCM/GA initialization has been compared with some other clustering approaches viz., (1) FCM modeling with random initialization. (2) PCM modeling with random initialization. (3) PCM modeling with FCM initialization. (4) PCM modeling with hybrid FCM/GA initialization. (5) PCM modeling with BCFCM initialization. (6) PCM modeling with hybrid BCFCM/GA initialization. (7) FPCM modeling with random initialization. (8) FPCM modeling with FCM initialization. (9) FPCM modeling with hybrid FCM/GA initialization. (10) FPCM modeling with BCFCM initialization. (11) FPCM modeling with hybrid GA/BCFCM initialization. (12) PFCM modeling with random initialization. (13) PFCM modeling with FCM initialization. (14) PFCM modeling with hybrid FCM/GA initialization. (15) PFCM modeling with BCFCM initialization. 

The Figure 8, Figure 9 and Figure 10 report TC measure based segmentation results for some images of datasets, by varying the additive noise level from 1% to 20%. 

In all performed experiments and for all simulated and real images, we noticed that modeling performance with BCFCM initialization outperformed that with FCM initialization. Noting further that the hybrid BCFCM/GA model produced the best results, convergence was fast with good compensation for noise. This is due to the regularization term integrated into the BCFCM algorithm and the benefit of optimization with GA.

In general, PFCM modeling with hybrid BCFCM/GA initialization was more accurate in providing the highest TC values compared to the results of other competing methods. However, we found that for near-low noise levels (from 1% to 5%), the performance of tissue segmentation was almost the same for all approaches. However, by increasing the noise level for values in the range [6%–20%], the results of the PFCM modeling with hybrid initialization BCFCM/GA are richly performing with higher quality of tissue volumes, with respect to the spatial inhomogeneities. The performance of the voxel classification was reduced by gradually increasing the noise level from 6% to 20%. Specifically, voxels that should be classified to GM tissue were falsely classified to WM and CSF tissues. While CSF voxels were falsely classified to GM and WM tissues.

The possible explanation for the success of the proposed BCFCM/GA/PFCM hybrid clustering scheme and for the good obtained results is that the FCM, PCM and FPCM approaches were more susceptible to the effect of noise, particularly FCM. In the result figures, this finding is clearly visible, where the FCM was unstable and sensitive to noise. The PCM gave balanced results, but the performance was insufficient. In contrast, hybrid models of fuzzy possibilistic clustering, FPCM and PFCM were stable and robust to noise with better performance, however, the PFCM was more efficient because, from 6% of noise, volumes became more difficult to segment, because of the accentuated noise and the spatial inhomogeneities that they contain. This requires a robust approach which adapts to the high noise level.

To ensure the performance of the proposed modeling approach, 20 experiments were performed with different random trials to obtain the BCFCM/GA initial partitions where the voxels were subjected to a noise level of 20%. For some images of ADNI, OASIS and GMPH dataset, Figure 11 and Figure 12 represent these TC and JS results obtained with the proposed modelling approach (BCFCM/GA/PFCM) (case *C* in the figures). The results of the two main PCM and FPCM clustering methods with a genetic BCFCM initialization (*A* and *B* in the figures) are reported for comparison purposes. These figures show the descriptive statistics of the data and the corresponding boxplots which allow giving the displaying of the distribution of data based on five-number summary the minimum, first quartile (Q1), median, third quartile (Q3), and maximum calculating the mean and the standard deviation (Sdev).

For all experiences and images, the same deductions were noticed. First, we can determine that PFCM’s median is higher (values in range [0.72,0.85]) than FPCM’s which is higher than PCM’s median. The PFCM has a small Sdev and in the majority of the experiences, the PFCM observations are split evenly at the median with a symmetric distribution, unlike for FPCM and PCM where the distribution of the measured data is skewed at the top or bottom.

The interquartile range (IQR = Q3 − Q1) for the proposed approach (*C*) is smaller in comparison with the FPCM and PCM approaches, but IQR-FPCM is smaller than IQR-PCM. This indicates that the TC and JS data for all experiences are more condensed (closer together). In order sense, the larger the IQR, the more variable the data set is (in the case of PCM, meaning the data are more spread out). 

Many suspected outliers were detected with the PCM, less in the case of FPCM and none with PFCM for all experiments (with the exception of a single outlier which was identified in experiment 3 for TC data and 17 for JS). In the charts, mild outliers (above Q3 + 1.5 IQR or bellow Q1 + 1.5 IQR) are expressed by an empty circle and extreme outliers (above Q3 + 3 IQR or bellow Q1 + 3 IQR) by a black circle.

### 4.3. Performance Evaluation of the CAD System

We used SE, SP, AC and AUC to validate the performance of SVDD. 

*SE* measures the percentage of sick people by AD who are correctly identified as having the condition. In another sense, it defines the proportion of actual positives that are correctly identified as such: (27)$$SE=\frac{\mathrm{number}\mathrm{of}\mathrm{TP}}{\mathrm{number}\mathrm{of}\mathrm{TP}+\mathrm{number}\mathrm{of}\mathrm{FN}}$$
where *TP* is the number of true positives which defines the number of AD patient volumes correctly classified, and *FN* is the number of false negatives which defines the number of AD patient volumes classified as control.

*SP* measures the percentage of healthy people who are correctly identified as not having the condition. In another sense, it defines the proportion of actual negatives that are correctly identified as such:(28)$$SP=\frac{\mathrm{number}\mathrm{of}\mathrm{TN}}{\mathrm{number}\mathrm{of}\mathrm{TN}+\mathrm{number}\mathrm{of}\mathrm{FP}}$$
where *TN* is the number of true negatives which defines the number of control volumes correctly classified, and *FP* is the number of false positives which defines the number of control volumes classified as AD patients’. As sensitivity and specificity reveal the ability to detect NOR/AD patterns, the best CAD system is the one that achieves the best trade-off between them.

CA is the proportion of correct predictions made by the CAD system. Formally, the CA is defined as:(29)$$CA=\frac{TN+TP}{TN+FP+TP+FN}$$

AUC characterizes the size of the area under the receiver operating characteristic (ROC) curve, which compares true positive rate (TPR) with false positive rate (FPR). In this case, the classifier is efficient if its AUC is close to 1. A ROC curve plots the TPR and FPR values for different classification thresholds, which are defined as follows:(30)$$TPR=\frac{TP}{TP+FN}$$
(31)$$FPR=\frac{FP}{FP+TN}$$

In each dataset, 70% of data were selected to form the classifier, and the remaining 30% of data to perform the tests. We opted for the cross-validation procedure to test the classifier’s performance and its robustness. Thus, the training and test sets were chosen 10 times at random to obtain at the end of the count 10 (70%–30%) training-test partitions. Figure 13 and Figure 14 and Table 2 provide the SVDD results of the MRI/PET multimodal classification in terms of AC, SE, SP and ROC curves.

Figure 15, Figure 16, Figure 17 and Figure 18 provide the corresponding boxplots of the descriptive statistics of the AC, SE, SP and AUC respectively, using 10-fold cross validation with 20% additive noise to evaluate the proposed SVDD classifier based on BCFCM/GA/PFCM segmentation approach (case *A* in figures) in comparison with SVM classifier, based on BCFCM/GA/PFCM (case *B* in figures). For all experiences and classification measures, we noticed that SVDD’s median is higher than SVM’s with small Sdev and IQR. This confirms that the values of measures are closer together.

Table 3 provides the results of the quantitative analysis based on the AC, SE, SP and AUC measurements of the proposed CAD system, including the results of several competing systems namely [5,46,47,54] for ADNI and [48] for OASIS. The proposed multimodal system provided the best results compared to those obtained by competing systems. The percentage of classification in terms of (AC, SE, SP, AUC) (%) was (85.09%, 86.14%, 84.92%, 94%) for GMPH dataset. In the case of the ADNI base, the result was (93.65%, 90.08%, 92.75%, 97.3%) for 159 multimodal MRI/PET images, compared with [5] which obtained (75%, 84.67%, 81.58%) using 95 MRI and (73%, 86.36%, 82.67%) using 95 PET images. Better results than those of [54] which acquired (89.64%, 87.10%, 92%, 94.45%) for 193 multimodal MRI/PET images. It was more efficient than the [46] system which achieved (81%, 78.52%, 81.76%) for 159 multimodal MRI/PET images. Added to this is the CAD system of [47], which got an accuracy rate of 89.52% using 105 PET images. For the OASIS database, the proposed multimodal system obtained (91.46%, 92%, 91.78%, 96.7%) with an equal error rate (EER) of 0.64% which is lower than the EER (0.72%) obtained in the work of [48] using the same conditions with 198 OASIS MR images. 

In addition, the proposed multimodal system was more efficient than its similar one using a single modality (MRI or PET). The results in the table highlight the advantages of multimodal fusion of brain images based on the one hand, on hybrid possibilistic-fuzzy-genetic segmentation and on the other hand, on the SVDD classifier, to clearly improve the performance of the classification system. In fact, all the values of the quantitative measurements obtained with the fusion of anatomical and functional images were superior to those of other competing CAD systems. In general, the best results were achieved when the fusion was based on the *FOP_4_* fusion operator.

With regard to computing time, the proposed CAD system with the use of the “divide and conquer” strategy converged more quickly. It consumed only 45 min for OASIS, 30 min for ADNI and 5 min for GMPH compared to its analogue without the application of this strategy and which consumed 2 h, 1 h, 45 min and 30 min for OASIS, ADNI and GMPH respectively.

## 5. Conclusions and Perspective

In this paper, a CAD system has been proposed to diagnose patients with probable prognosis for AD. It was mainly based on two modules. First, a segmentation module that takes into consideration noise, associated PVE, inhomogeneity of intensity, and weak boundaries that limit the diagnostic potential of the MRI and PET brain images. In this context, a multimodal fusion process was proposed based on the fuzzy-genetic-possibilistic foundations to model uncertain and inaccurate data in medical images. It is composed of three stages, modeling, fusion and decision. We first derive fuzzy tissue maps by modeling the degree of relationship between a voxel and a given tissue. These maps are then combined into fused maps using a possibilistic conjunctive operator highlighting redundancies in fuzzy maps and managing ambiguous areas but relevant for diagnosis. The interest of the fusion is then demonstrated by a labeling process and by the synthesis of a new high resolution functional image which exploits the anatomical information for the precise detection of pathological zones.

We first adopt BCFCM to determine the initial partition of tissue classes. Afterwards, the problem of fuzzy clustering was posed as an optimization problem, applying the GA to obtain the most optimal initial partition. Then, a modeling based on the PFCM is carried out in order to model the degree of relationship between each tissue class and a given voxel. This hybrid possibilistic-fuzzy algorithm overcomes some weaknesses of its analogues, namely FCM, PCM and FPCM. It solves the FCM noise sensitivity deficit problem, overcomes the problem of coincident PCM clusters, and provides an improvement to the FPCM by eliminating the sum constraints of the FPCM lines. 

We then propose as a second module, an automatic learning approach based on the one-class SVDD discriminative classifier, which allows distinguishing between the brain images belonging to patients with AD and those of patients in normal aging. The advantage of such a classifier is its robustness against outlier points. All the obtained experimental results show that this kind of classifier represents a promising approach in the presence of incomplete and imprecise training knowledge, allowing a flexible adaptation of the classification architecture to the available information.

The performance of the proposed CAD system has been validated on several sets of synthetic and real brain images. For the proposed tissue quantification approach, results of some slices chosen because of their tissue particularities, are given. In general, the hybrid fuzzy-genetic-possibilistic fusion process allowed segmentation of acceptable quality and superior performance to some state-of-the-art methods for a 20% noise level. 

To ensure the superiority of the CAD system based on the SVDD classifier, a comparison has been made with other proposed systems. The higher classification percentage in the case of the proposed system compared to the performance of other proposed classifiers in the literature, demonstrates the interest of the SVDD classifier and the superior capabilities of the multimodal fusion approach in comparison to the consideration of a single modality in the segmentation of cerebral images.

The results of the CAD system obtained for patients suffering from AD were encouraging; however, many perspectives are expected for this prospective study. The application of approaches for T1-weighted anatomical images could be remedied using T2 weighting, proton density and other types of MRI images. It is also desirable to look for other larger databases that have noisy brain images and approximate to actual clinical images. It is also interesting to adapt the proposed approaches to detect other brain disorders such as Huntington’s disease, Parkinson’s disease, etc.

Our current concern was to provide for the expert radiologist labeling images and a functional synthesis image which exploits the anatomical information for the precise detection of pathological zones. According to some studies, the hippocampus is probably the first of these pathological brain areas affected by the pathogenic mechanisms of AD. Our study touches on some of these aspects but unfortunately, the distribution of hippocampal cell loss in normal aging and in AD is not well understood at this time. In this context, an extended study will be developed in the near future.

Further work is needed to improve the quality of the CAD system. As a result, as a perspective of this work, other more robust hybrid algorithms will be desired for segmentation, and other classifier models with explanatory reasoning will be desired to model the complex and incomplete data.

## Figures and Tables

**Figure 1 brainsci-09-00289-f001:**
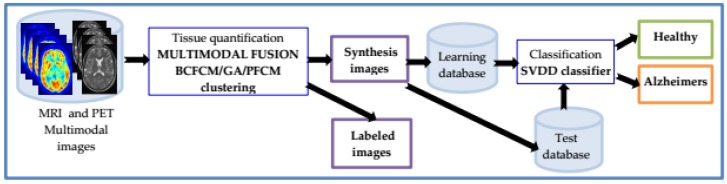
Overall block diagram of the proposed multimodal CAD system for early diagnosis of AD. Based on hybrid fuzzy-possibilistic-genetic tissue quantification and SVDD classifier.

**Figure 2 brainsci-09-00289-f002:**
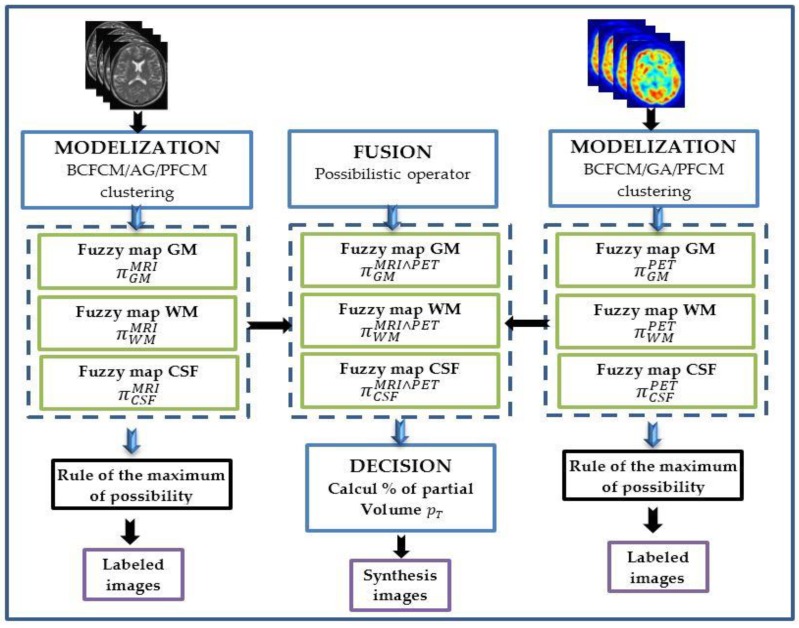
Proposed general scheme for tissue quantification based on multimodal MRI/PET fusion based on modeling, fusion and decision steps.

**Figure 3 brainsci-09-00289-f003:**
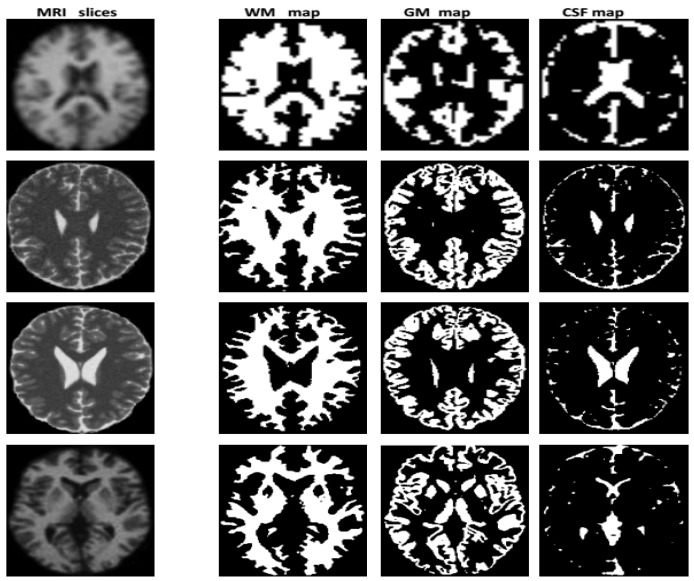
Fuzzy maps of WM, GM and CSF tissues calculated with PFCM modeling based on hybrid BCFCM/GA centers initialization, with additive noise level is 20%, slice thickness is 5 and inhomogeneity of the radio frequency is 20%.

**Figure 4 brainsci-09-00289-f004:**
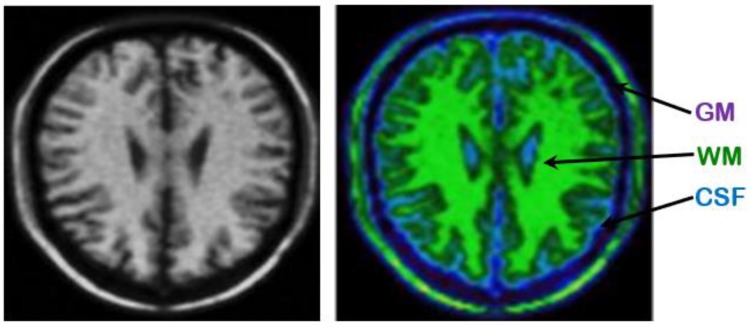
Example of a labeled anatomical image from automatic segmentation based on the hybrid GA/BCFCM/PFCM clustering approach with additive noise level is 20%, slice thickness is 5 and inhomogeneity of the radio frequency is 20%.

**Figure 5 brainsci-09-00289-f005:**
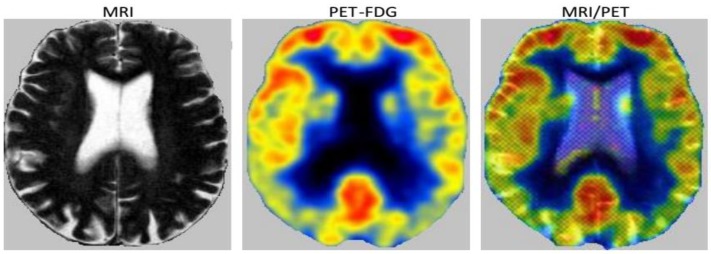
Example of anatomical and functional images of the same brain and the synthetic image resulting from the automatic segmentation based on the proposed multimodal fusion approach.

**Figure 6 brainsci-09-00289-f006:**
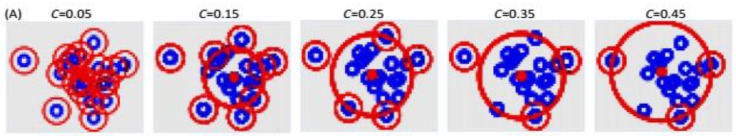
Example of SVDD solutions with: σ = 0 and different values of *C*, the circled points represent the support vectors.

**Figure 7 brainsci-09-00289-f007:**
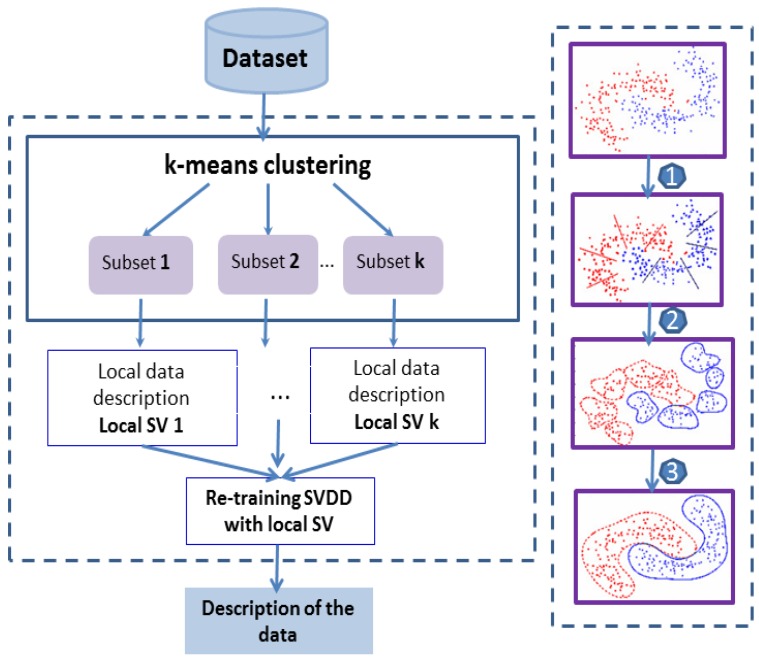
Outline of “divide and conquer” strategy for the division and training process based on the k-means clustering and SVDD sub-classifiers trained with only local support vectors (SV).

**Figure 8 brainsci-09-00289-f008:**
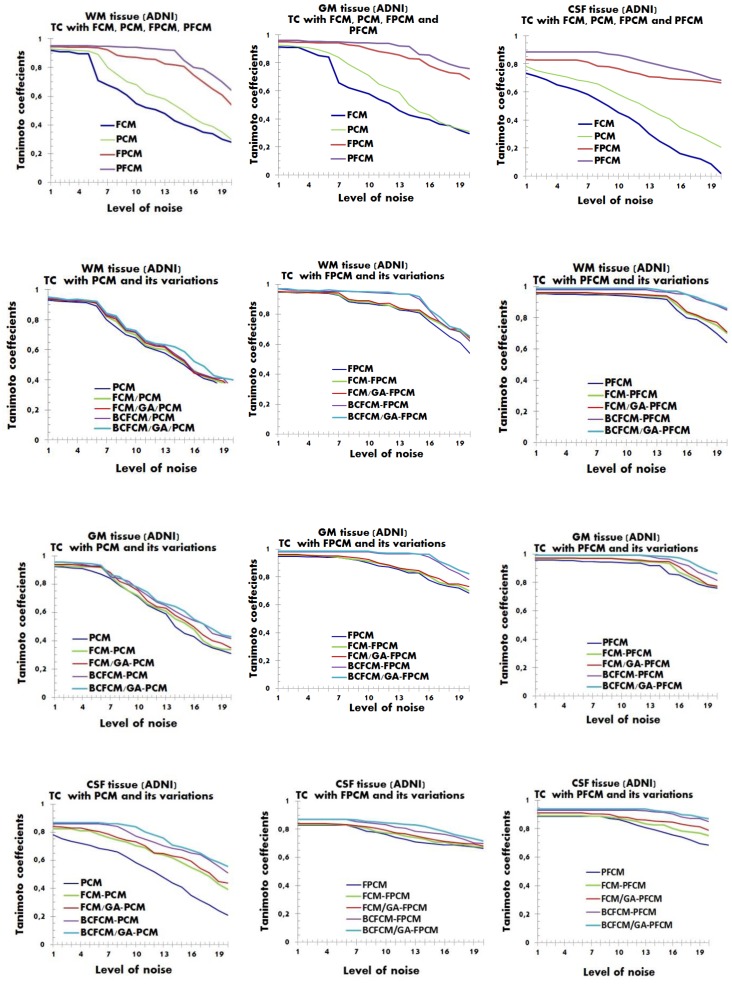
Curve of Tanimoto coefficients clustering index for WM, GM and CSF tissues with additive noise between 1% and 20%. Application with the hybrid approach BCFCM/GA/PFCM for an ADNI image. The results of modeling approaches from (1) to (15) are reported viz., (1) FCM, (2) PCM, (3) FCM/PCM, (4) FCM/GA/PCM, (5) BCFCM/PCM, (6) BCFCM/GA/PCM, (7) FPCM, (8) FCM/FPCM, (9) FCM/GA/FPCM, (10) BCFCM/FPCM, (11) BCFCM/GA/FPCM, (12) PFCM, (13) FCM/PFCM, (14) FCM/GA/PFCM, (15) BCFCM /PFCM. **Legend:** BCFCM: Bias corrected FCM; FCM: Fuzzy c-means; GA: Genetic algorithms; PCM: Possibilistic c-means; FPCM: Fuzzy PCM; PFCM: Possibilistic FCM.

**Figure 9 brainsci-09-00289-f009:**
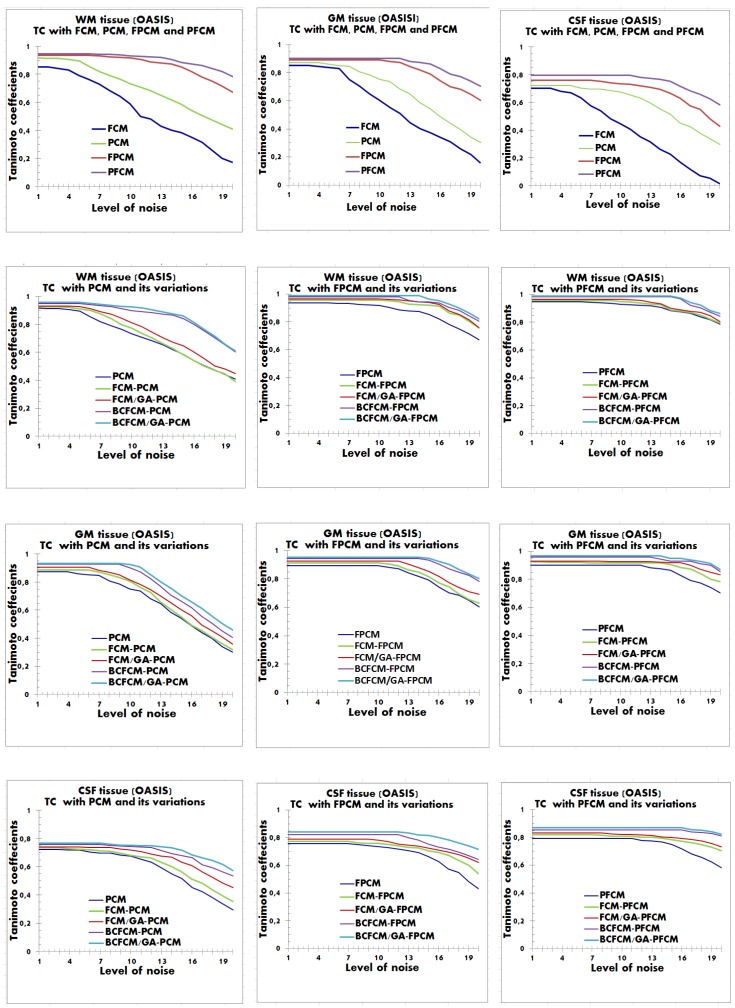
Curve of Tanimoto coefficients clustering index for WM, GM and CSF tissues with additive noise between 1% and 20%. Application with the hybrid approach BCFCM / GA / PFCM for an OASIS image. The results of modeling approaches from (1) to (15) are reported viz., (1) FCM, (2) PCM, (3) FCM/PCM, (4) FCM/GA/PCM, (5) BCFCM/PCM, (6) BCFCM/GA/PCM, (7) FPCM, (8) FCM/FPCM, (9) FCM/GA/FPCM, (10) BCFCM/FPCM, (11) BCFCM/GA/FPCM, (12) PFCM, (13) FCM/PFCM, (14) FCM/GA/PFCM, (15) BCFCM /PFCM. **Legend**: BCFCM: Bias corrected FCM; FCM: Fuzzy c-means; GA: Genetic algorithms; PCM: Possibilistic c-means; FPCM: Fuzzy PCM; PFCM: Possibilistic FCM.

**Figure 10 brainsci-09-00289-f010:**
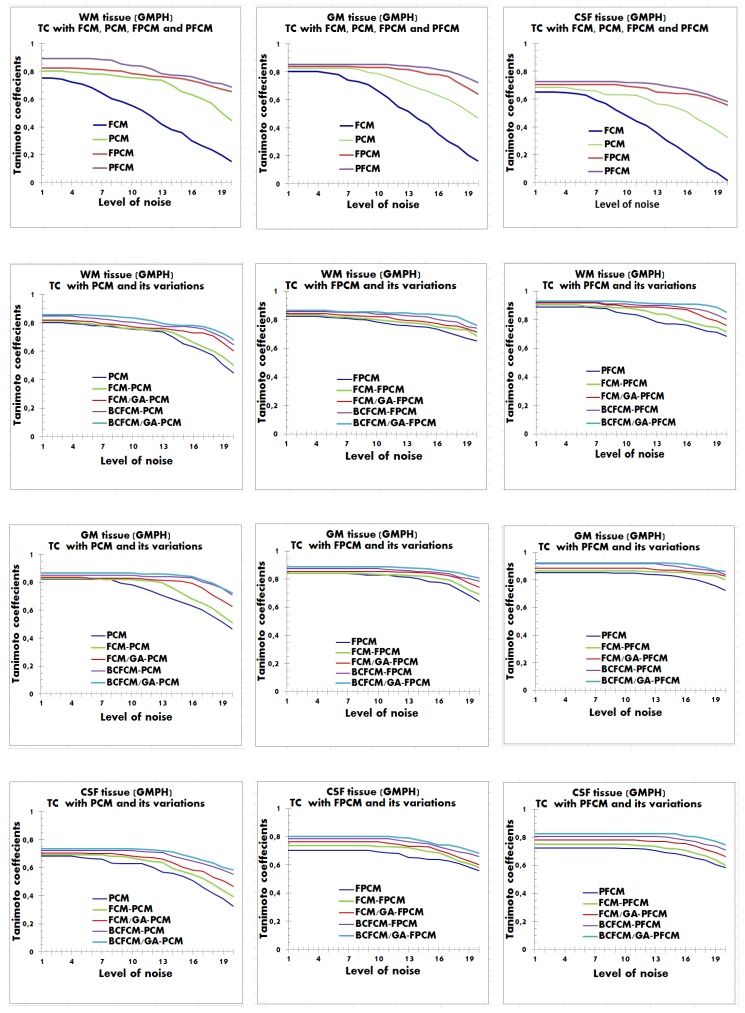
Curve of Tanimoto coefficients clustering index for WM, GM and CSF tissues with additive noise between 1% and 20%. Application with the hybrid approach BCFCM/GA/PFCM for a GMPH image. The results of modeling approaches from (1) to (15) are reported. viz., (1) FCM, (2) PCM, (3) FCM/PCM, (4) FCM/GA/PCM, (5) BCFCM/PCM, (6) BCFCM/GA/PCM, (7) FPCM, (8) FCM/FPCM, (9) FCM/GA/FPCM, (10) BCFCM/FPCM, (11) BCFCM/GA/FPCM, (12) PFCM, (13) FCM/PFCM, (14) FCM/GA/PFCM, (15) BCFCM /PFCM. **Legend.** BCFCM: Bias corrected FCM; FCM: Fuzzy c-means; GA: Genetic algorithms; PCM: Possibilistic c-means; FPCM: Fuzzy PCM; PFCM: Possibilistic FCM.

**Figure 11 brainsci-09-00289-f011:**
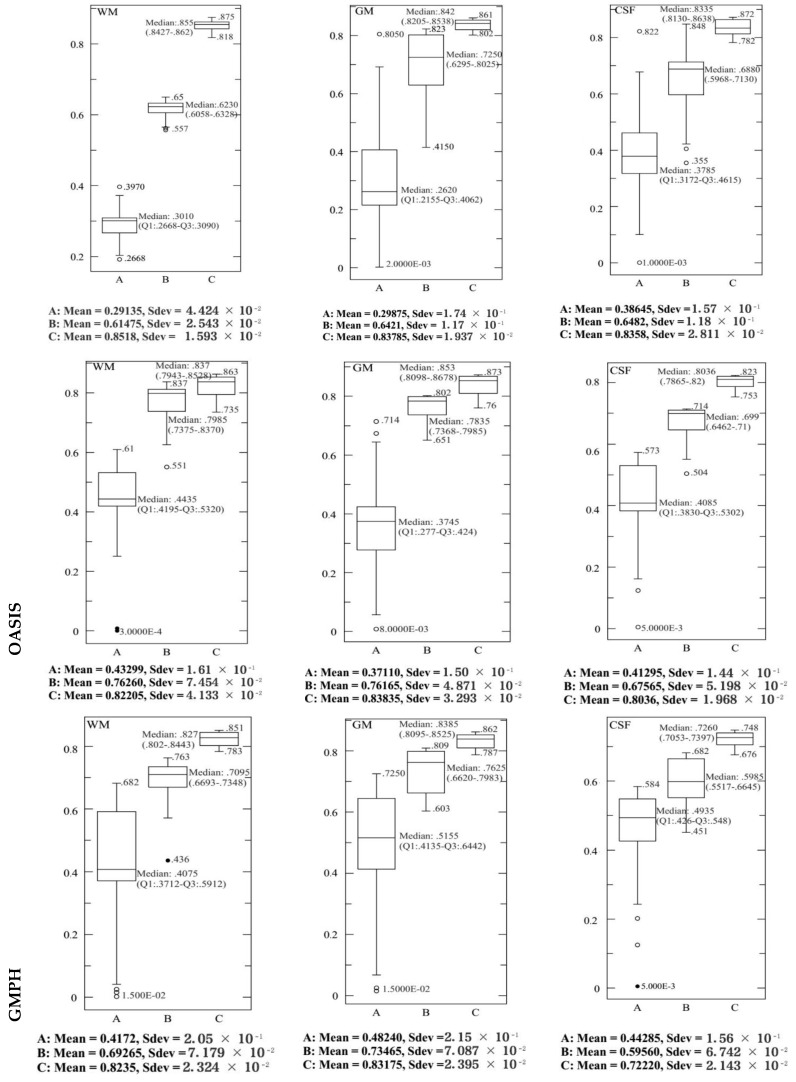
The corresponding Box-and-whisker plots of the descriptive statistics of the measured TC values (95% confidence interval) for WM, GM and CSF tissues with 20% additive noise. 20 experiences are used to evaluate the following modelling approaches: BCFCM/GA/PFCM (*C*), BCFCM/GA/FPCM (*B*) and BCFCM/GA/PCM (*A*).

**Figure 12 brainsci-09-00289-f012:**
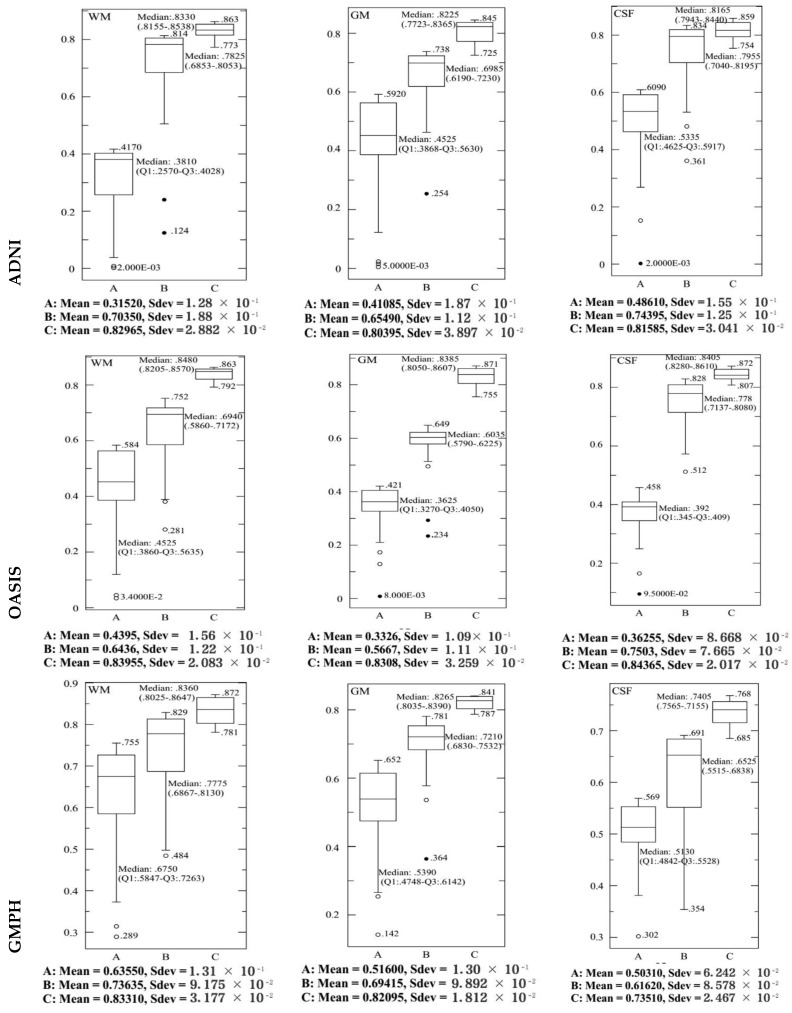
The corresponding Box-and-whisker plots of the descriptive statistics of the measured JS values (95% confidence interval) for WM, GM and CSF tissues with 20% additive noise. 20 experiences are used to evaluate the following modelling approaches: BCFCM/GA/PFCM (*C*), BCFCM/GA/FPCM (*B*) and BCFCM/GA/PCM (*A*).

**Figure 13 brainsci-09-00289-f013:**
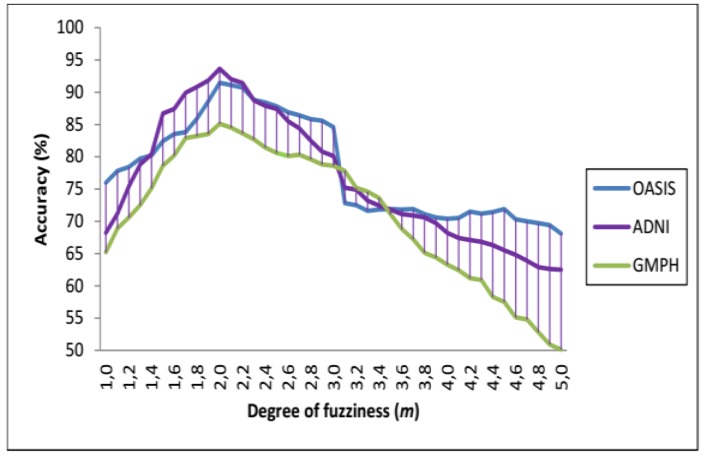
Accuracy results of the MRI/PET multimodal classification with SVDD (RBF function with *σ* = 0.5) for the ADNI, OASIS, GMPH databases during the variation of the fuzzy parameter *m* in the range [1, 5] with a step of 0.1.

**Figure 14 brainsci-09-00289-f014:**
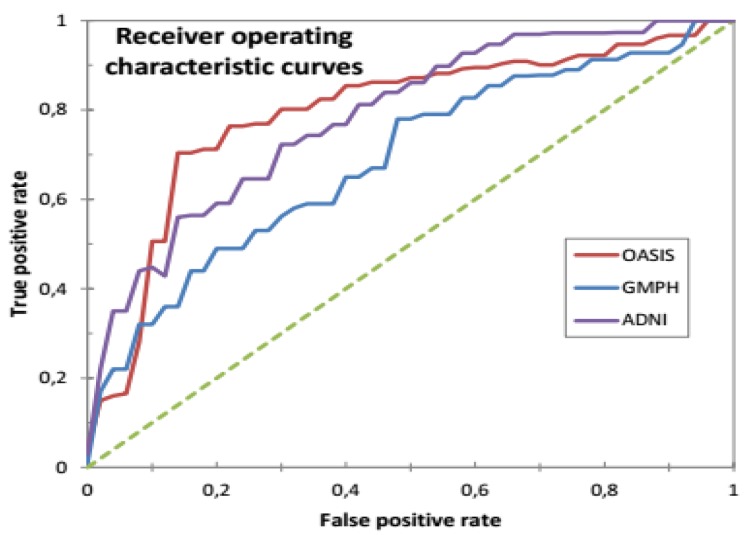
ROC curves for classification of AD vs. HC with SVDD (RBF function with σ = 0.5) for MRI /PET images from ADNI, OASIS and GMPH databases.

**Figure 15 brainsci-09-00289-f015:**
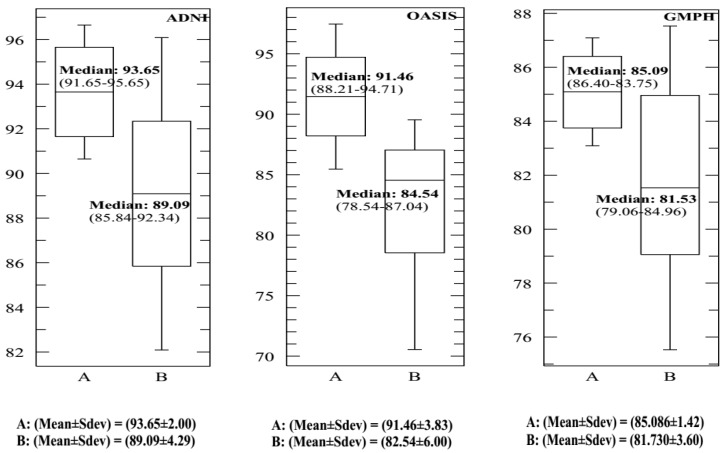
The corresponding Box-and-whisker plots of the descriptive statistics of the accuracy (AC) values for ADNI, OASIS and GMPH datasets with 20% additive noise, using 10-fold cross validation to evaluate the proposed SVDD classifier based on BCFCM/GA/PFCM segmentation approach (case *A* in figure). Comparison with SVM classifier, based on BCFCM/GA/PFCM (case *B*).

**Figure 16 brainsci-09-00289-f016:**
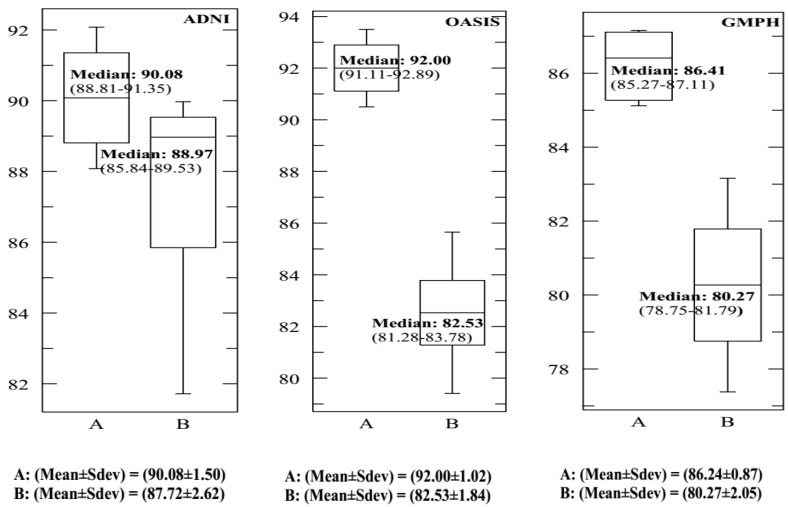
The corresponding Box-and-whisker plots of the descriptive statistics of the sensitivity (SE) values for ADNI, OASIS and GMPH datasets with 20% additive noise, using 10-fold cross validation to evaluate the proposed SVDD classifier based on BCFCM/GA/PFCM segmentation approach (case *A* in figure). Comparison with SVM classifier, based on BCFCM/GA/PFCM (case *B*).

**Figure 17 brainsci-09-00289-f017:**
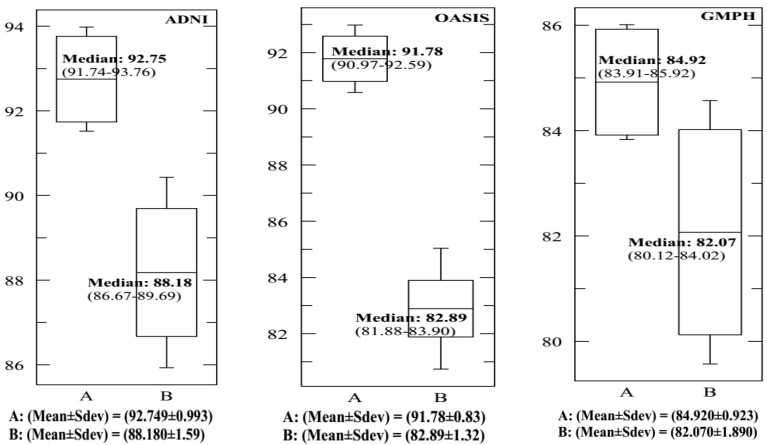
The corresponding Box-and-whisker plots of the descriptive statistics of the specificity (SP) values for ADNI, OASIS and GMPH datasets with 20% additive noise, using 10-fold cross-validation to evaluate the proposed SVDD classifier based on BCFCM/GA/PFCM segmentation approach (case *A* in figure). Comparison with SVM classifier, based on BCFCM/GA/PFCM (case *B*).

**Figure 18 brainsci-09-00289-f018:**
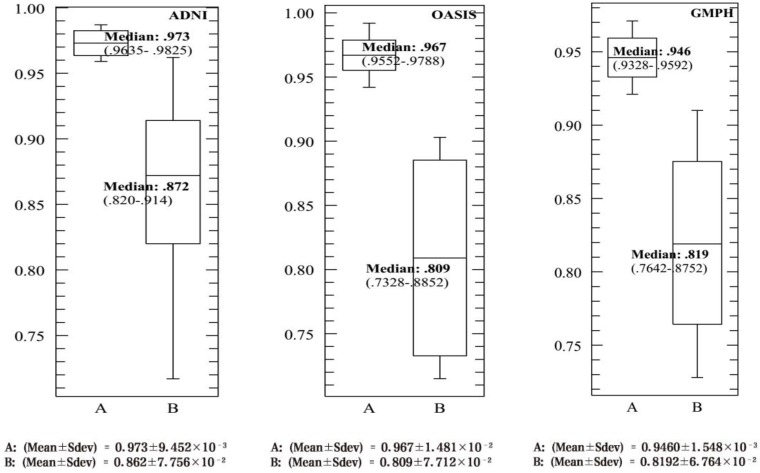
The corresponding Box-and-whisker plots of the descriptive statistics of the area under ROC curve (AUC) values for ADNI, OASIS and GMPH datasets with 20% additive noise, using 10 fold cross validation to evaluate the proposed SVDD classifier based on BCFCM/GA/PFCM segmentation approach (case *A* in figure). Comparison with SVM classifier, based on BCFCM/GA/PFCM (case *B*).

**Table 1 brainsci-09-00289-t001:** Demographic information of the participants and the characteristics of the magnetic resonance imaging (MRI) and positron emission tomography (PET) images.

**Characteristics of Patients**						
	**ADNI** [60]		**OASIS** [61]		**GMPH**	
	**AD**	**Healthy**	**AD**	**Healthy**	**AD**	**Healthy**
**Nb. patients**	77	82	100	98	5	-
**Woman/Man**	42/35	42/40	104/94		2/3	-
**Age**	75.4 ± 7.1	75.3 ± 5.2	70.17 (42.5–91.7)		71–86	-
**Education**	14.9 ± 3.4	15.6 ± 3.2	15.2 ± 2.7 (8–23)		-	-
**MMSE (base)**	23.8 ± 1.9	29.0 ± 1.2	29.1 ± 0.8 (27–30)		-	-
**MMSE (2 years)**	19.3 ± 5.6	29.0 ± 1.3	-	-	-	-
**ADAS-Cog (b)**	18.3 ± 6.1	7.3 ± 3.3	-	-	-	-
**ADAS-Cog (2 y)**	27.3 ± 11.7	6.3 ± 3.5	-	-	-	-
**Characteristics of the images (MRI/PET)**						
	**ADNI**		**OASIS**		**GMPH**	
	**AD**	**Healthy**	**AD**	**Healthy**	**AD**	**Healthy**
**RF (%)**	20	20	20	20	20	-
**ST (mm)**	1, 3, 5	1, 3, 5	1, 3, 5	1, 3, 5	1, 3, 5	-
**SNR (%)**	1–20	1–20	1–20	1–20	1–20	-
**Nb. slices**	20	20	4	4	64	-
**Nb. volumes**	60	60	60	60	60	-
**Nb. total**	92,400	98,400	24,000	23,520	19,200	-

**Legend**: MMSE = Mini-Mental State Examination, ADAS-Cog = Alzheimer’s Disease Assessment Scale-Cognitive subscale, SNR = additive noise, ST = slice thickness, RF= Radio frequency, b = base, y = year.

**Table 2 brainsci-09-00289-t002:** Performance results of support vector data description (SVDD) classifier in terms of accuracy (AC)%, sensitivity(SE)% and specificity (SP)% using 10-fold cross-validation and grid search to perform (σ, *C*) hyper-parameter optimization of SVDD.

**ADNI (AC%, SE%, SP%)**									
**(σ, *C*)**	***C* = 0.004**	***C* = 0.05**	***C* = 0.00625**	***C* = 0.125**	***C* = 0.15**	***C* = 0.25**	***C* = 0.35**	***C* = 0.45**	***C* = 0.5**
**σ =** 0	90.15 89.21 81.04	89.24 88.39 80.99	89.01 87.74 80.12	88.41 87.13 79.59	88.05 86.75 79.03	87.65 86.36 78.51	87.04 86.04 78.08	86.38 85.72 77.38	86.02 85.07 77.52
**σ =** 0.5	**93.65** **91.46** **85.09**	93.15 91.03 84.64	93.01 89.57 84.03	92.47 89.07 83.56	92.29 88.46 83.29	92.03 88.03 83.01	91.83 87.86 82.46	91.42 87.53 82.39	91.05 87.02 82.00
**OASIS (AC%, SE%, SP%)**									
**σ =** 0	88.54 88.35 87.00	88.35 88.14 86.75	88.02 88.02 86.24	87.61 87.75 86.00	87.34 87.25 85.61	87.01 87 85.23	86.82 86.84 84.99	86.41 86.36 84.76	86.04 86.14 84.51
**σ =** 0.5	90.34 91.35 91.06	90.86 91.01 91.35	**91.46** **92.00** **91.78**	90.06 90.65 90.54	89.48 90.24 90.00	89.02 90.03 87.89	89 89.82 87.26	88.89 89.07 87.03	88.55 89 87.00
**GMPH (AC%, SE%, SP%)**									
**σ =** 0	85.09 86.14 84.92	85.09 86.14 84.92	85.09 86.14 84.92	85.09 86.14 84.92	85.09 86.14 84.92	85.09 86.14 84.92	85.09 86.14 84.92	85.09 86.14 84.92	85.09 86.14 84.92
**σ =** 0.5	84.58 86.00 84.07	**85.09** **86.14** **84.92**	83.85 85.65 83.53	83.02 85.07 83.08	84.36 84.62 82.14	83.84 84.02 81.73	83.19 83.72 81.27	83 83.38 80.74	82.47 83.07 80.44

**Legend.***C*: Parameter that controls the tradeoff between volume of a hypersphere and the number of errors (penalty parameter), σ: width parameter of Gaussian RBF kernel function.

**Table 3 brainsci-09-00289-t003:** Performance of the proposed computer-aided diagnosis (CAD) system for early diagnosis of AD using 10-fold cross validation. Comparison with some systems referenced in the literature.

Reference	Classification	Segmentation	Data	Modality	Nb. Patients	SNR (%)	AC (%)	SE (%)	SP (%)	AUC	EER (%)
This study	SVDD (RBF) 10-FCV	BCFCM/GA /PFCM	ADNI	MRI/PET	159 (77 AD, 82 HC)	20	**93.65**	**90.08**	**92.75**	**0.9730**	-
This study	SVDD (RBF) 10-FCD	BCFCM/GA /PFCM	ADNI	MRI	159 (77 AD, 82 HC)	20	88.15	89.02	90.18	0.9500	-
This study	SVDD (RBF) 10-FCV	BCFCM/GA /PFCM	ADNI	PET	159 (77 AD, 82 HC)	20	85.16	86.84	84.14	0.9204	-
This study	SVM (RBF) 10-FCV	BCFCM/GA /PFCM	ADNI	MRI/PET	159 (77 AD, 82 HC)	20	89.09	87.72	88.18	0.8720	-
This study	SVM (RBF) 10-FCD	BCFCM/GA /PFCM	ADNI	MRI	159 (77 AD, 82 HC)	20	83.42	82.34	87.51	-	-
This study	SVM (RBF) 10-FCV	BCFCM/GA /PFCM	ADNI	PET	159 (77 AD, 82 HC)	20	80.72	84.34	81.23	-	-
[5]	SVM (RBF) LCV	FCM/PCM	ADNI	MRI	95 (45 AD, 50 HC)	20	75	84.67	81.58	-	-
[5]	SVM(RBF) LCV	FCM/PCM	ADNI	PET	95 (45 AD, 50 HC)	20	73	86.36	82.67	-	-
[54]	CNN 10-FCV	NA	ADNI	MRI/PET	193(93 AD, 100 HC)	NA	89.64	87.1	92	0.9445	-
[47]	SVM (RBF)	PCA/LDA	ADNI	PET	105 (53 AD, 52 HC)	NA	89.52	-	-	-	-
[47]	FFNN	PCA/LDA	ADNI	PET	105 (53 AD, 52 HC)	NA	88.75	-	-	-	-
[46]	MKL-SVM 10-FCV	NA	ADNI	MRI/PET	159 (77 AD, 82 HC)	NA	81	78.52	81.76	0.885	-
This study	SVDD (RBF) 10-FCV	BCFCM/GA /PFCM	OASIS	MRI/PET	198 (100 AD, 98 HC)	20	**91.46**	**92.00**	**91.78**	**0.9670**	**64**
This study	SVDD (RBF) 10-FCV	BCFCM/GA/ PFCM	OASIS	MRI	198 (100 AD, 98 HC)	20	81.46	78.57	83.73	0.9041	-
This study	SVDD (RBF) 10-FCV	BCFCM/GA /PFCM	OASIS	PET	198 (100 AD, 98 HC)	20	79.24	80.16	83.58	0.8538	-
This study	SVM(RBF) 10-FCV	BCFCM/GA /PFCM	OASIS	MRI/PET	198 (100 AD, 98 HC)	20	82.54	82.53	82.89	0.8090	-
This study	SVM(RBF) 10-FCV	BCFCM/GA /PFCM	OASIS	MRI	198 (100 AD, 98 HC)	20	76.52	74.21	79.49	-	-
This study	SVM(RBF) 10-FCV	BCFCM/GA /PFCM	OASIS	PET	198 (100 AD, 98 HC)	20	74.82	74.27	81.86	-	-
[48]	SVM(linear) LCV	k-means/GA	OASIS	MRI	198 (100 AD, 98 HC)	NA	-	-	-	-	72
This study	SVDD (RBF) 10-FCV	BCFCM/GA/ PFCM	GMPH	MRI/PET	5 (5 AD, 0 HC)	20	**85.09**	**86.41**	**84.92**	**0.946**	-
This study	SVDD (RBF) 10-FCV	BCFCM/GA /PFCM	GMPH	MRI	5 (5 AD, 0 HC)	20	76.82	83.43	82.98	0.8608	-
This study	SVDD (RBF) 10-FCV	BCFCM/GA /PFCM	GMPH	PET	5 (5 AD, 0 HC)	20	73.49	81.07	75.71	0.8089	-
This study	SVM(RBF) 10-FCV	BCFCM/GA /PFCM	GMPH	MRI/PET	5 (5 AD, 0 HC)	20	81.53	80.27	82.07	0.8190	-
This study	SVM(RBF) 10-FCV	BCFCM/GA /PFCM	GMPH	MRI	5 (5 AD, 0 HC)	20	75.01	80.21	76.48	-	-
This study	SVM(RBF) 10-FCV	BCFCM/GA /PFCM	GMPH	PET	5 (5 AD, 0 HC)	20	71.72	78.73	71.92	-	-

**Legend:** BCFCM: Bias corrected FCM; CNN: Convolutional neural networks; FCM: Fuzzy c-means; FCV: Fold cross-validation; FFNN: Feed-forward neural network; GA: Genetic algorithms; LCV: Leave-one-out cross-validation; LDA: Linear Discriminant Analysis; MKL: Multiple Kernel Learning; PCA: Principal Component Analysis; PCM: Possibilistic c-means; PFCM: Possibilistic FCM; SVDD: Support vector data description; SVM: Support vector machines.

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
