# Peer review of "Computer-Aided Diagnosis System of Alzheimer’s Disease Based on Multimodal Fusion: Tissue Quantification Based on the Hybrid Fuzzy-Genetic-Possibilistic Model and Discriminative Classification Based on the SVDD Model"

_brainsci, 2019, doi:10.3390/brainsci9100289_

Round 1
Reviewer 1 Report
Thanks to authors for correction. The current revised version seems good enough for publication except abstract. Author should describe the method briefly in the abstract- detail description making is messy.
Reviewer 2 Report
Despite paper is a bit hard to follow and winding at times, I must admit that authors have one way or another addressed (at least in part) most of my previously raised concerns, and I believe manuscript has been now improved, at least providing further statistical validation of results shown.
This manuscript is a resubmission of an earlier submission. The following is a list of the peer review reports and author responses from that submission.
Round 1
Reviewer 1 Report
This paper deals with a CAD system for the automatic diagnosis of Alzheimer Disease (AD) based on fuzzy-genetic (fuzzy-clustering) and support vector machine (SVM) approaches from MRI and PET images. My major concerns has to do with the validation of results, which I believe is at least limited and could be improved, where some of the results shown look preliminary to me, as shown next in no particular order:
MAJOR CONCERNS:
1- The study, despite looking sound, does not provide statistical valid results after randomized trials. I suggest including repeated random trials to at least incorporate mean /pm standard deviation values where appropriate (even more so considering the addition of noise to samples), instead of only single scalar values. If possible, the inclusion of boxplot statistical dispersion figure results would be very desirable.
2- In an automatic classification system (CAD), the usual way to measure performance include: the correct classification rate, confusion matrices and Receiver Operation Characteristic (ROC) curves, with Area Under Curve (AUC) analysis, specially for binary classification problems (AD versus HC). I suggest to extend results to take into account ROC and AUC, like for instance:
https://towardsdatascience.com/understanding-auc-roc-curve-68b2303cc9c5
https://en.wikipedia.org/wiki/Receiver_operating_characteristic
3- Conclusions: it is my understanding that various conclusion statements, need to be toned down in present results as shown in manuscript:
3.1 "We first derive fuzzy tissue maps by modeling the degree of relationship between a voxel and a given tissue. These maps are then combined into fused maps using a possibilistic conjunctive operator for a more robust representation. The interest of the fusion is then demonstrated by a labeling process and by the synthesis of a new high resolution functional image."
The more robust nature of the representation must be properly proven or shown in manuscript numerical statistical valid results. Otherwise, I suggest to tone down the previous statement, with another kind of more mild rewritting, like " for an extended representation" or something similar.
3.2- Ibidem for the following paragraphs inside conclusions (need to be toned down):
"Several visual and quantitative results on multimodal images demonstrate
that the hybrid fuzzy- genetic-possibilistic fusion process allowed for good quality segmentation and outperformed state-of-the-art methods for high noise levels"
3.3 Ibidem for: please rewrite and tone down
"Several visual and quantitative results on multimodal images demonstrate
that the hybrid fuzzy- genetic-possibilistic fusion process allowed for good quality segmentationand outperformed state-of-the-art methods for high noise levels"
Minor comments:
a- "level of noise" should be formally/properly defined in manuscript (c.f. Figs. 8, 9 and 10).
b- In general terms, figure and table captions should be improved to include more detail about the results shown in them. For instance, the acronyms shown in Table 2, could be include inside the table caption, to avoid the reader to go and search for those through out the entire manuscript.
c- Table 2: some cases included show a number of 0 HC. Please, further explanations are needed to understand what kind of classification is being made with only one class samples (AD).
d- English usage must be double checked. For instance, table 1 includes French language in it.
e- It is not always clear whether supervised (RBF) or unsupervised (c-means clustering) learning approaches are being used. Please clarify where appropriate.
f- Learning function (objective mathematical function and eqs.) equations (1)-(11) may be considered to be included in an ad-oc appendix section of manuscript (I am only wondering).
g- The visual quality of figures 8, 9 and 10, might be improved.
h- Reference list is in general bit old. Please try also to include some relevant recent references in the field.
Reviewer 2 Report
This topic of this article is interesting and worth to investigate. It introduces multimodal images (MR and PET) analysis based method for Alzheimer disease diagnosis in order to combine both stuctural and functional information of brain images. this article compares it outcome with some existing methods and obtained satisfactory outcome. However, the length and organization of the draft is too broad and untidy- my first impression is to reduce redundency (unnecessary texts/info) of the draft.
Major comments:
Abstract is too broad. Pls reduce main themes including challenges, contribution and achievements.
Section 1 to 4. Background of the study is too long. Pls describe the problems, challenges of the study and existing solutons and your contribution more especifically.
How multimodal registration was performed- need more explanation.
How segmentation result was evaluated,
Some study suggested that hippocampus is the first target to diagnosis algheimer disease. How this study consider this matter.
Please mention some similar approaches where machine learning based classifier was employed in brain disease diagnosis after extracting features. Bellow there are some examples.
M. Yasugi, B. Hossain, M. Nii, and S. Kobashi, “Relationship Between Cerebral Aneurysm Development and Cerebral Artery Shape,” J. Adv. Comput. Intell. Intell. Inform., Vol.22, No.2, pp. 249-255, 2018. Z. Watanabe, et al., “Comparison of Rates of Growth between Unruptured and Ruptured Aneurysms Using Magnetic Resonance Angiography,” J. of Stroke and Cerebrovascular Diseases, Vol.26, No.12, pp. 2849-2854, 2017.
Minor comments:
Lines 21, meaning is obscure , "Fusion using a possibilistic operator to merge the maps of the two 21 types of anatomical and functional images. -"
Mention the difference of the two images in Fig .5.
Changes to english name. Title of the table 1 "Caractéristiques des patients"
LIne 146, pls write down location of 'Clermont ferrand'
Discussion under the section 2 is not required. it should merge with literature review.
Title of "3. Discussion" should be different.
Round 2
Reviewer 1 Report
Despite authors have provided Tanimoto coefficients graphs at various noise levels for the evaluation of results, and have addressed in part several suggestions and concerns I raised, I am sorry to say that unfortunately authors have not taken into account my first major concern which limits the proper statistical validation of numerical results shown, and time-delay reasons for not providing them are given.
Author Response
"Please see the attachment. Please see the comments whose answer is in purple. All paragraphs written in purple, represent possible improvements from your comments for the second round"

Reviewer 2 Report
The authors should remove redundant text, most of the reader are aware of the benefit of the fusion of multimodal images, rather author should mention the benefit of both modalities, i.e high spatial resolution of MRI combined with the functional information from PET,
1 abstract --> pls delete
"The collection of various data, from various modalities as well as specialized 13 knowledge is becoming more and more common in clinical departments for the study of Alzheimer's 14 disease (AD). Unfortunately, the development of analysis tools capable of understanding the large 15 amount of generated data as well as the complexity of the brain image structures remains to be done."
SVDD- > Support Vector Data Description (SVDD)
FIg. 1. databasis --> database
Pls use english only in figure 2.
Line 701, How common coordinate system is defined in each type of image? need further explanation in a figure.
Fig. 6. How the fusion image MRI/PET is synthesized. Any tool is used?. Pls use 3D plot in the fig.
To find out the optimal value of sigma and C in SVDD, its common to use grid search method. Pls use grid search and then compare superiority of your chosen value over the standard method.
pls mention, How so many reference labeled images were collected- how many experts. how operators bias was minimized. Segmentation of WM, GM, CSF is very important in this study because it could significantly bias the classification.
Overall, the authors should go through the draft very carefully for minor correction. The length of the draft is broad. Reader might be confused and miss important information due to its length.
Author Response
"Please see the attachment. Please see in the paper, the comments whose answer is in purple. All paragraphs written in purple, represent possible improvements for your comments for the second round."
